# Alkalinity responses to climate warming destabilise the Earth's thermostat

Nele Lehmann [1,2,3] ✉, Tobias Stacke [1,8], Sebastian Lehmann[4], Hugues Lantuit [2,5], John Gosse [6], Chantal Mears [1], Jens Hartmann [7] & Helmuth Thomas [1,3] ✉

Alkalinity generation from rock weathering modulates Earth's climate at geological time scales. Although lithology is thought to dominantly control alkalinity generation globally, the role of other first-order controls appears elusive. Particularly challenging remains the discrimination of climatic and erosional influences. Based on global observations, here we uncover the role of erosion rate in governing riverine alkalinity, accompanied by areal proportion of carbonate, mean annual temperature, catchment area, and soil regolith thickness. We show that the weathering flux to the ocean will be significantly altered by climate warming as early as 2100, by up to 68% depending on the environmental conditions, constituting a sudden feedback of ocean $CO_2$ sequestration to climate. Interestingly, warming under a low-emissions scenario will reduce terrestrial alkalinity flux from mid-latitudes ($-1.6$ t(bicarbonate) $a^{-1}$ $km^{-2}$) until the end of the century, resulting in a reduction in $CO_2$ sequestration, but an increase ($+0.5$ t(bicarbonate) $a^{-1}$ $km^{-2}$) from mid-latitudes is likely under a high-emissions scenario, yielding an additional $CO_2$ sink.

Weathering-derived alkalinity fluxes to the ocean are a key component of the Earth's carbon cycle[1]. Weathering of both carbonate and silicate rocks consumes atmospheric/soil $CO_2$ and increases the alkalinity of the ocean. Carbonate and silicate weathering is thought to be in equilibrium with marine calcification at long timescales (-10 ka)[2], yet it has the potential to alter alkalinity, and thus the carbon cycle, over millennial and shorter timescales. Beyond the calcium carbonate compensation time, only silicate weathering acts as a long-term sink for atmospheric $CO_2$, while carbonate weathering acts $CO_2$-neutral.[1]

Compared to silicate rocks, carbonate rocks are weathered more rapidly, and their dissolution kinetics are up to three orders of magnitude faster[3,4]. This allows carbonate weathering to be more responsive to rapid environmental changes, like acid rain and anthropogenic increases in groundwater $CO_2$ levels[5,6]. Environmental changes occurring over shorter timescales than calcium carbonate compensation

(<10 ka)[2], e.g., changes in regional temperature, hydrology, vegetation, and atmospheric $CO_2$ content, have the potential to shift carbonate weathering away from its steady state, thereby altering the Earth's carbon cycle[7]. It is therefore essential to assess the short-term and long-term anthropogenic, climatic, and geologic drivers of global alkalinity fluxes. Terrestrial river discharge and watershed lithology are recognized as the two most dominant factors controlling global alkalinity fluxes to the ocean[8–12]. Acidity, which is mostly supplied from atmospheric and soil $CO_2$, and buffered by the rate of physical erosion, also exerts first-order controls on global alkalinity fluxes. The influence of temperature at the global scale has been long debated[8,12], and two recent global studies strongly indicate optimal weathering in temperate climates[13,14]. However, precisely quantifying these controls (e.g., soil $CO_2$ content) at regional to global scales remains intangible, and records that are both spatially diverse and temporally comprehensive

[1]Institute of Carbon Cycles, Helmholtz-Zentrum Hereon, Geesthacht, Germany. [2]Alfred Wegener Institute Helmholtz Centre for Polar and Marine Research, Potsdam, Germany. [3]Institute for Chemistry and Biology of the Marine Environment (ICBM), University of Oldenburg, Oldenburg, Germany. [4]Buchholz in der Nordheide, Germany. [5]Institute of Geosciences, University of Potsdam, Potsdam, Germany. [6]Department of Earth and Environmental Sciences, Dalhousie University, Halifax, NS, Canada. [7]Institute for Geology, Center for Earth System Research and Sustainability (CEN), University Hamburg, Hamburg, Germany. [8]Present address: Max Planck Institute for Meteorology, Hamburg, Germany. ✉e-mail: nele.lehmann@hereon.de; helmuth.thomas@hereon.de

remain scarce[7,14]. While regional studies show that physical erosion can enhance weathering, at least for low-to-moderate erosion rates (supply limitation)[15–19], no such relationship between physical erosion and alkalinity flux has been identified at the global scale[8,11].

Here, we combine riverine alkalinity measurements with in situ [10]Be-derived erosion rates from multi-lithological catchments across different climate zones to reveal the impact of physical erosion on alkalinity generation in the global context. We then quantify the impact of two future climate scenarios, represented by two shared socio-economic pathways (SSPs), on the riverine alkalinity flux.

## Results and discussion
### First-order controls on riverine alkalinity
We compiled data from 233 sampling locations on six continents ranging from 44°S to 51°N, for which both alkalinity and [10]Be erosion rate measurements were available (Fig. 1a). To overcome the ramifications of runoff and alkalinity concentration (i.e., dilution by "pure" water or evaporation), we use runoff-normalized alkalinity as the ratio of observed alkalinity in a given river sample to the mean annual runoff of that river. We sought to characterize alkalinity in a volume-independent manner and therefore considered alkalinity

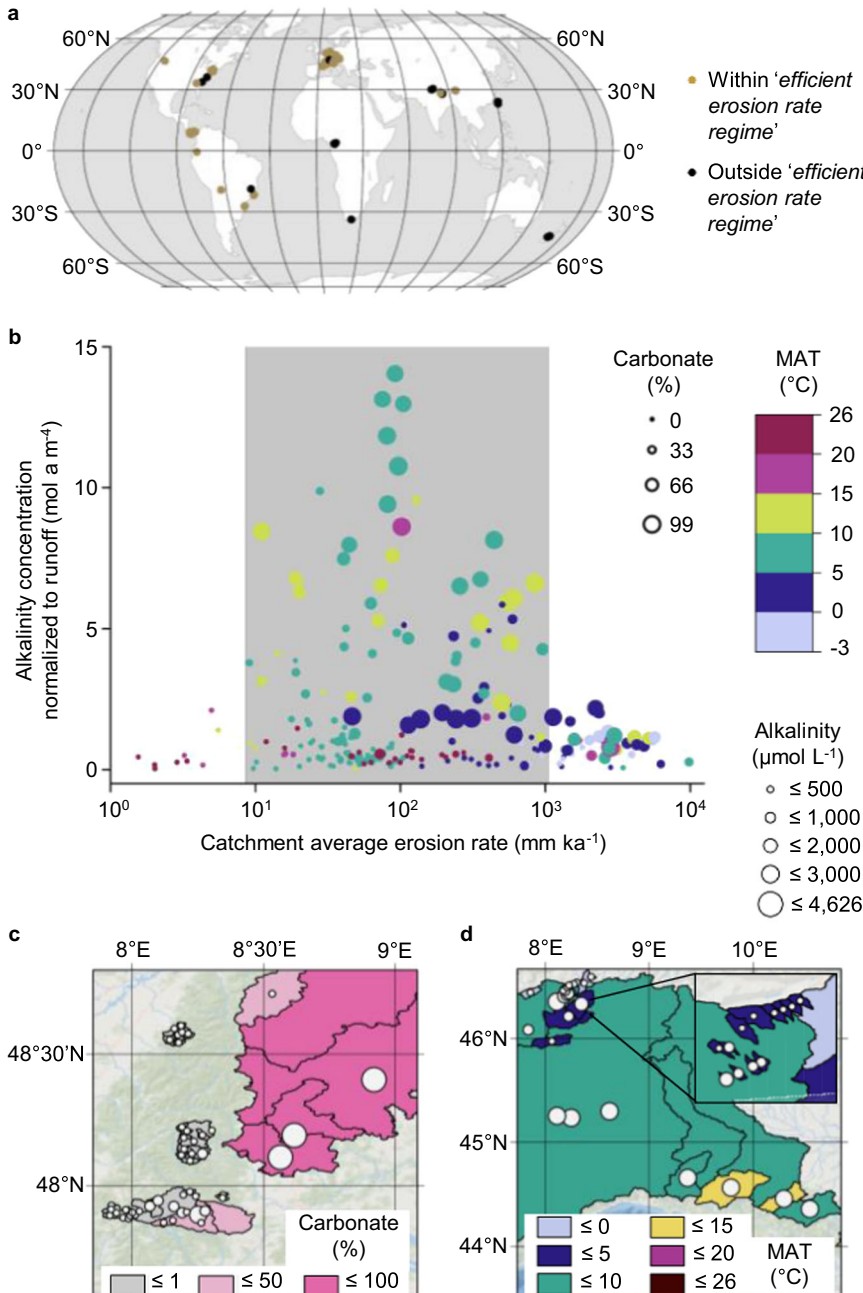

**Fig. 1 | Erosion rate, areal carbonate proportion, and temperature are first-order controls on catchment-scale alkalinity concentrations. a** World map with sampling locations. Catchments within the limits of the *efficient erosion rate regime*, characterized by erosion rates of -10–1000 mm ka[-1] (see **b**), are highlighted. **b** High runoff-normalized alkalinity concentrations are found in the *efficient erosion rate regime* (gray-shaded box). A high proportion of areal carbonate and a temperate climate (MAT: 5–15 °C) promote high-alkalinity concentrations, as shown in the excerpts for selected European catchments: **c** Black Forest, Germany, where a high areal proportion of carbonate is associated with high riverine alkalinity; and **d** Switzerland and northern Italy, where catchments with MATs of 5–15 °C show high-alkalinity concentrations. MAT: mean annual temperature. The background map in (**a**) was created in ArcGIS Pro with data from Living Atlas, Natural Earth, and Esri's country and water shapes[67]. The background maps in (**c**) and (**d**) are from Esri, DeLorme, Natural Vue, and GEBCO[67].

concentration per unit runoff. In the following, we will first highlight the relationships of erosion rate, areal carbonate proportion and mean annual temperature (MAT) with runoff-normalized alkalinity that we can derive from our dataset. Then we will show the results of how our empirically-based modeling approach represents these relationships. We employed a generalized linear model (GLM), allowing us to establish linear relationships between runoff-normalized alkalinity and the various predictor variables, even though their actual relationship may not be linear. This allowed for a readily understandable interpretation. In addition to erosion rate, areal carbonate proportion, and MAT, our model analyses normalized alkalinity as a function of catchment area and soil regolith thickness (see "Methods").

Our model results show that erosion rate is a first-order control on riverine alkalinity at the global level. Catchments with both low ($<10$ mm ka$^{-1}$) and high ($>1000$ mm ka$^{-1}$) erosion rates produce little alkalinity, independent of areal carbonate proportion and MAT. Where erosion rates are very slow, i.e., less than ~10 mm ka$^{-1}$, we suggest that the stream dissolved load is too low for alkalinity production. Similarly, catchments that erode rapidly ($>1000$ mm ka$^{-1}$) produce little alkalinity because they appear to be equilibrium-limited, i.e., limited by acid availability (see below for further explanation). At slow to intermediate erosion rates (~10–1000 mm ka$^{-1}$, gray-shaded box, Fig. 1b), we identify a regime of "*efficient erosion rate*" which induces the highest normalized alkalinity concentrations. A study in tropical Taiwan[20] confirms the increase in carbonate weathering as a function of erosion rate for an erosional gradient similar to our study. However, discrepancies appear at high erosion rates, where we obtain decreasing alkalinities. We attribute this to a high degree of pyrite weathering in the studied area in Taiwan, according to the authors, which would result in higher groundwater acidity and thus higher solubility of carbonates in their study.

Within the *efficient erosion rate regime*, normalized alkalinity peaks at erosion rates of ~100 mm ka$^{-1}$ and is governed by the areal proportion of carbonate and MAT in the catchment. Our global dataset indicates that areal carbonate proportion has a first-order positive effect on normalized alkalinity concentration. The highest normalized alkalinity concentrations are only found in catchments with carbonate present, while in catchments without carbonate, weathering produces only low amounts of alkalinity (Fig. 1b, c and Supplementary Fig. 1). Our model results show that weathering from carbonate rocks dominates alkalinity generation globally. This predominance of carbonate was also recognized by another study[9], in which the flux of CO$_2$ consumed by weathering in carbonate watersheds was determined to be 17 times higher than in plutonic and metamorphic watersheds, which reveal the lowest flux. Our results also indicate that, when other influential parameters (e.g., erosion rate and MAT) are ideally set (i.e., within the *efficient erosion rate regime*, ~10–1000 mm ka$^{-1}$, and MAT ~10 °C), catchments dominated by rock types other than carbonate (areal carbonate proportion ≤50%) still produce high amounts of alkalinity (normalized alkalinity concentration ~2.5–10.0 mol a m$^{-4}$, third-quartile of normalized alkalinity concentration and alkalinity concentration in our dataset ~2.1 mol a m$^{-4}$ and ~2010 µmol L$^{-1}$, respectively; Fig. 1b and Supplementary Fig. 1).

In addition to erosion rate and areal carbonate proportion, we identify MAT as a first-order control on riverine alkalinity globally. A temperate climate (5–15 °C) promotes extensive carbonate weathering (Fig. 1b, d). This is supported by our model, where normalized alkalinity concentration shows a maximum in the range of 7.5–15.0 °C (Supplementary Fig. 2b). Similar conclusions have been drawn by other authors[13,14], who found the highest carbonate weathering rates to be in temperate climates. They[14] describe the temperature dependency of alkalinity by a Gaussian function, with an optimum at 11 °C. Furthermore, they attribute the increase in alkalinity concentration

until 11 °C MAT to an increase in soil–rock CO$_2$ content supplied by elevated ecosystem respiration. Above 11 °C, the effect of decreased carbonate mineral solubility with increasing temperature causes alkalinity concentration to decrease again. Our results confirm that temperature, in addition to influencing kinetics and solubility, is an important driver of acid availability.

Colder MATs ($<5$ °C) generally hinder weathering and are associated with low normalized alkalinity concentrations, even when both the areal carbonate proportion is high ($>80\%$) and the erosion rate lies within the *efficient erosion rate regime* (Fig. 1b, d). We attribute this to reduced ecosystem respiration and, hence, to lower acid availability. In addition, weathering may be limited at low temperatures due to reduced reaction rates according to the Arrhenius equation. In contrast, higher MATs ($>15$ °C) can produce high normalized alkalinity concentrations, but only in watersheds that are both within the *efficient erosion rate regime* and have a high areal carbonate proportion. However, these conditions are rarely encountered, since warmer climate zones are normally associated with only a small areal carbonate proportion (Supplementary Fig. 3). This is observed in our global dataset, where the vast majority of catchments with higher MAT ($>15$ °C) have a relatively low areal carbonate proportion (Fig. 1b). We propose that normalized alkalinity concentration decreases at MATs greater than ~12.5 °C and reaches its minimum at ~22.5 °C. We relate the decreasing trend in alkalinity to the generally semi-arid conditions associated with this temperature regime globally. This is supported by prior work[21] demonstrating that water availability limits chemical and physical weathering processes in dry environmental conditions. Overall, we suggest that the MAT covariate in our model is representative of the availability of soil acid and water, and that the ideal conditions for weathering-liberated alkalinity are met in temperate catchments with high areal carbonate proportion, within the *efficient erosion rate regime*.

Beyond peak alkalinity, at intermediate to high erosion rates ($>100$ mm ka$^{-1}$), a general decrease in acid availability in both ground and river water correlates with a decrease in riverine alkalinity, consistent with prior work in Western Europe[4,7]. In regions in the Jura mountains undergoing extensive carbonate weathering, Calmels et al.[7] found that soil CO$_2$ content decreases with altitude. The authors propose that this is linked to a change in vegetation (above 800 m above mean sea level (amsl)), climate, and soil properties. Indeed, our global dataset reveals, first, a positive correlation between erosion rate and altitude ($R^2 = 0.5$) and, second, that catchments with a mean elevation of at least 1500 m amsl are characterized by an erosion rate $>100$ mm ka$^{-1}$ (Supplementary Fig. 4a). There is also a positive correlation between erosion rate and mean slope gradient, which is representative of relief ($R^2 = 0.7$; Supplementary Fig. 4b). This suggests that denudation causes soil CO$_2$ to decrease with elevation and relief. Taking a different approach, Erlanger et al.[4] found that in the Northern Apennine Mountains of Italy, precipitation of secondary calcium carbonate from supersaturated rivers led to the loss of 20–90% of dissolved Ca$^{2+}$ from carbonate-rich catchments. The carbonate super-saturation is a result of degassing of excess river CO$_2$ through equilibration with atmospheric CO$_2$ partial pressure. Turbulent flow aids this degassing process. We propose that a reduction in riverine normalized alkalinity concentration may be found in the rapidly eroding catchments in our dataset because they are associated with steep slopes (as mentioned above, erosion rate and mean slope gradient are positively correlated), which in turn are responsible for turbulent flow in rivers. It seems that morphology, and not directly erosion rate, is mainly responsible for turbulent flow and thus CO$_2$ degassing. However, morphology and erosion rate are closely linked. In catchments with high erosion rates, extreme flow events can transport large boulders into the riverbed, which in turn cause turbulent flow.

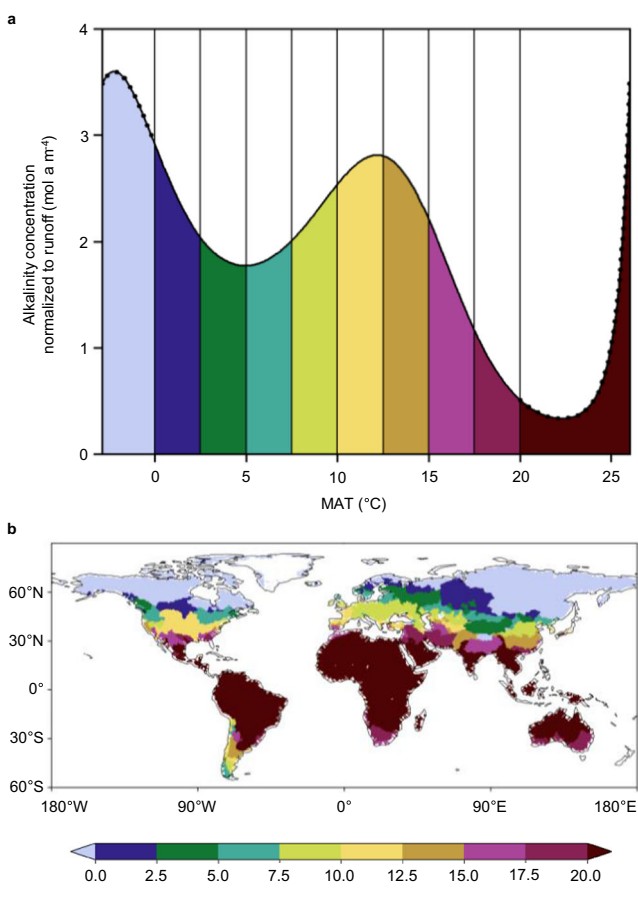

**Fig. 2 | Runoff-normalized alkalinity concentration in catchments with different MAT. a** The fitted line shows the model output as a function of MAT (mean annual temperature); the line is dotted for MAT <0 °C and ≥20 °C, indicating that these temperatures were not included into the quantitative interpretation of the influence of MAT on normalized alkalinity concentration. The other covariates are kept constant (mean global coverage by carbonates (sc + sm) = 22%[50], erosion rate = 100 mm ka$^{-1}$, catchment area = 1000 km², soil regolith thickness = 15 m). **b** MAT of catchments globally, derived from data provided by the ISIMIP project[63] based on the GFDL-ESM4 Model[68]. See also Supplementary Fig. 2b. The background map with the continent outlines in (**b**) is from naturalearthdata.com[69].

## Global riverine alkalinity function

From an iterative model and a variable selection process, we developed a generalized linear model to describe normalized alkalinity concentration. In addition to erosion rate, areal carbonate proportion, and MAT, we included catchment area, calculated as 2D area, and soil regolith thickness (depth to bedrock)[22] as covariates (these five covariates are also significant predictors for alkalinity flux when included in a generalized additive model; see "Methods"). Normalized alkalinity concentration increases continuously for both catchment area and soil regolith thickness (Supplementary Fig. 2c, d). First, it is possible that larger catchment areas, which tend to capture more precipitation, provide a greater power for lateral planation or incision. In fact, the stream power law as derived by Moss et al.[23] is commonly expressed as erosion rate $E = kA^m S^n$, which substitutes catchment area, A, for water discharge $Q_w$ ($Q_w = kA^m$, c.f. ref. [24]). However, the stream power law applies more to the stream processes of incision that generate suspended, bed, and dissolved loads rather than to processes that generate only dissolved solids. As riverine alkalinity corresponds directly to the dissolved load, there are likely other factors that can manifest

spatial scaling. A larger watershed may also provide opportunities to include areas of steeper slopes, giving more energy for incision and slope wash (the S in the above stream power law). However, larger catchments have a higher potential for sediment storage and greater land surface areas of low relief (e.g., ref. [25]) which provide a greater opportunity for weathering and soil development. It is this increase in chemical weathering on flatter and finer-grained unconsolidated landforms that likely contributes a first-order link between dissolved load and catchment area. Other authors[8] found that alluvial sediments in small, steep Japanese basins are relatively poorly weathered in comparison to larger basins in other regions, possibly where sediment storage and shallower slopes increase weathering efficiency. Second, beyond peak weathering rates at an optimum soil regolith thickness, chemical weathering is thought to decrease due to an ineffective interaction between water and fresh mineral surfaces[26]. In wet regions, chemical weathering is usually extensive, but the leached bedrock has not been physically moved downslope, and soil production rates are often slow[27]. Soil production is generally driven by stochastic bioturbation (e.g., penetration of roots), which produces well-mixed and mobile soil layers[28,29]. The disturbance frequency is governed by soil thickness, and soil production rates are thought to generally decrease non-linearly with increasing soil thickness[30,31]. We acknowledge the divergent feedbacks between chemical weathering, biotic processes, and soil production and transport processes. However, we cannot observe this trend in our dataset (Supplementary Fig. 5). Except for the four data points with the highest soil regolith thicknesses (>21 m) in our dataset, which show low normalized alkalinity concentrations, we detect a continuous increase of normalized alkalinity concentration with increasing soil regolith thickness. In future studies, it would be beneficial to include catchments with higher soil regolith thicknesses (>25 m) to investigate whether decreasing weathering can be detected above a certain soil regolith thickness.

## Sensitivity of alkalinity generation to MAT

We employed our model to assess how MAT affects normalized alkalinity concentrations in different climatic zones (Fig. 2). For this sensitivity test, we applied global mean values for the remaining covariates: erosion rate, areal carbonate proportion, catchment area and soil regolith thickness. As shown in Supplementary Fig. 2b, the expression of the normalized alkalinity–MAT function is (partly) altered by varying the values of the other covariates. The general dependency of the function, as shown in Fig. 2a, however, is not changed.

For the temperature range of 0–20 °C, MAT exerts a strong to intermediate influence on normalized alkalinity concentration in our model. At high latitude, in subarctic or alpine climates (<2.5 °C), our model predicts an increase in normalized alkalinity concentration with decreasing MAT. We relate this enhanced carbonate weathering to glacial and periglacial erosion. A study investigating the geochemistry of rivers draining the New Zealand Alps[32] found that glaciated watersheds contain ~25% more Sr²⁺ from carbonate weathering than non-glaciated ones. In our dataset, normalized alkalinity increases with the areal extent of permanent snow and ice cover (for all catchments with permanent snow and ice cover >1%; Supplementary Fig. 6). This correlation can be explained by enhanced weathering at the margins of the glaciers, where meltwaters interact with atmospheric $CO_2$ and fine-grained glacial debris[33]. For non-glaciated catchments with an MAT around 5 °C at ~60–45°N, normalized alkalinity concentration is relatively small owing to lower acid availability (annual soil respiration decreases with latitude[34]). In temperate climates (-12.5 °C) at ~45–35°N, both water and soil acid are abundant, resulting in an increase in normalized alkalinity concentration. In (semi-)arid climates (>15 °C), at ~35–25°N, weathering is limited by water supply, resulting in the lowest

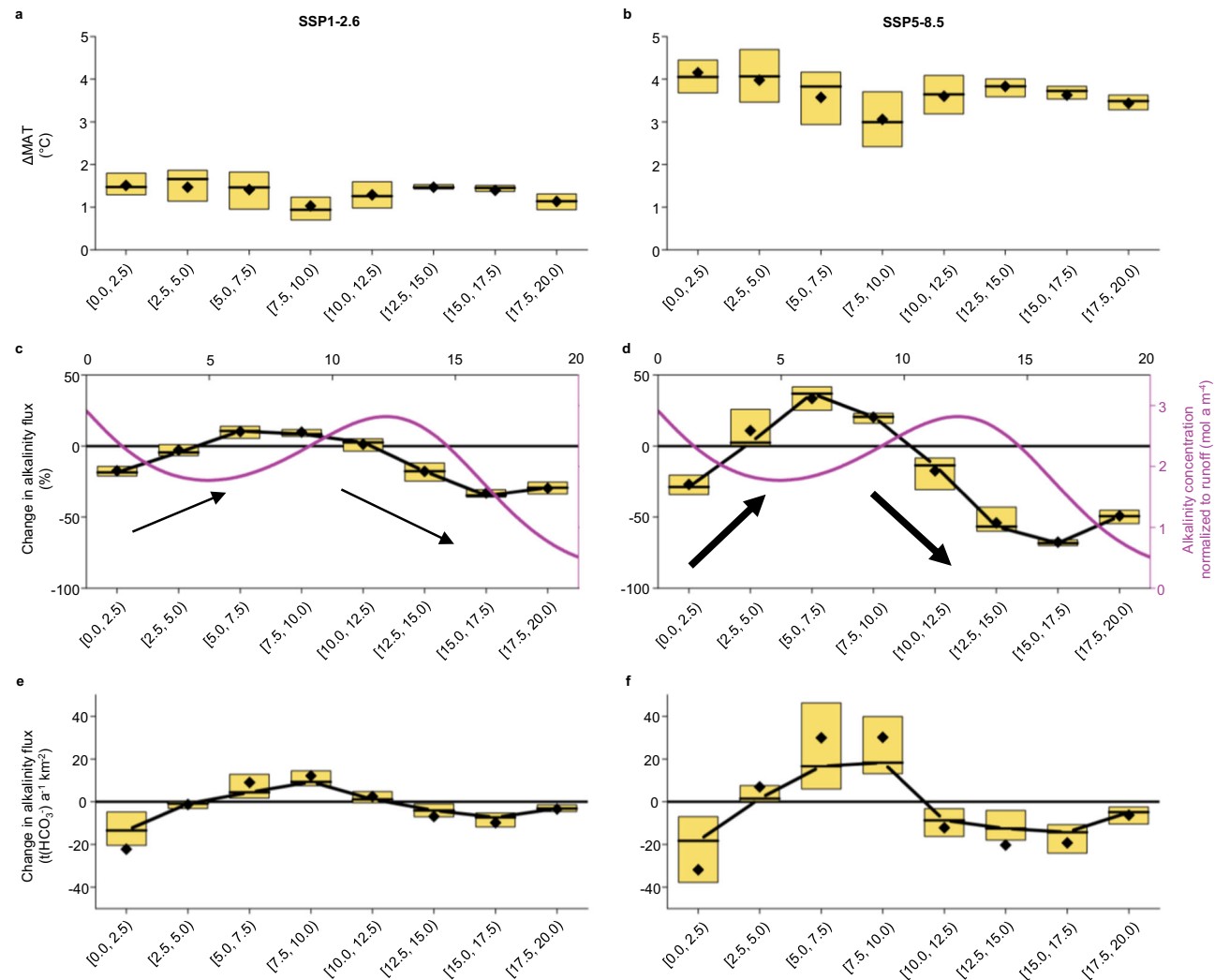

**Fig. 3 | Alkalinity flux impacted by increased temperatures.** Simulated historical data (1980–2009) are contrasted with simulated future data affected by climate warming according to (**a, c, e**) a low (SSP1-2.6) emissions scenario and (**b, d, f**) a high (SSP5-8.5) emissions scenario (2070–2099). **a, b** Difference in MAT; **c, d** Relative change in alkalinity flux due to change in MAT. The schematic evolution of runoff normalized alkalinity concentration according to our model (from Fig. 2a) is shown for a better understanding. Thick arrows indicate that weathering responds more drastically to more rapidly rising temperatures than to less rapidly rising temperatures, indicated by the thin arrows; and **e, f** Absolute change in alkalinity flux

due to change in MAT. For the calculation of the absolute alkalinity flux as specific mass flux, a molar mass of 61.02 g mol⁻¹ for bicarbonate ($HCO_3^-$) was used, as at pH 7–9, the alkalinity concentration is approximately equal to the bicarbonate concentration[45,46]. Boxes indicate 0.25 and 0.75 quantiles, and black diamonds show the arithmetic mean. Temperature projections are provided by the ISIMIP project[63] based on the GFDL-ESM4 data[68]. River discharge was simulated using the HydroPy global hydrology model[65]. SSP shared socioeconomic pathway, MAT mean annual temperature.

normalized alkalinity concentration. Finally, our model predicts an increase in normalized alkalinity concentration for warmer climates (>22.5 °C). We attribute this increase to elevated reaction rates according to the Arrhenius law.

Our model predicts high and very high normalized alkalinity concentrations for polar (<0 °C) and tropical (>25 °C) regions, respectively (dotted line in Fig. 2a). While our dataset covers a broad temperature range [−2.9, 26.9 °C], we restrict our quantitative interpretation of the influence of MAT on normalized alkalinity concentration to a more narrow temperature range of 0.0–20.0 °C. Outside this range, (i) we would perform an out-of-sample prediction regarding the areal carbonate proportion, and (ii) the implicit constancy of the erosion rate covariate (erosion rate = 100 mm ka⁻¹) appears unrealistic for these regions. Our global dataset reveals that catchments with high MAT (>20 °C) have a low areal carbonate proportion (mean = 2.8%, Supplementary Fig. 7). In fact, our dataset does not contain any catchments that show both high temperatures

(>20 °C) and high areal carbonate proportion (>40%). However, some catchments in warm areas (>20 °C) to which we apply our model function in our global assessment also have high carbonate contents (up to 100%). For the catchments with low MAT (< 0 °C) in our dataset, the assumption of an erosion rate of 100 mm ka⁻¹ is untenable, as the mean erosion rate for this temperature regime is ~2600 mm ka⁻¹, with almost all observations falling above the *efficient erosion rate regime* (Supplementary Fig. 8). Therefore, in the following, we will limit our quantitative assessment of the effect of MAT changes on alkalinity to the temperature range of 0.0–20.0 °C.

**Impact of climate change on the terrestrial alkalinity flux**
Global surface temperatures are projected to increase over the next decades under all emissions scenarios[35] (shown for two shared socioeconomic pathways (SSPs); a low (SSP1-2.6) and a high (SSP5-8.5) emissions scenario in Fig. 3a, b, respectively). SSPs are future narratives that combine projections of atmospheric greenhouse gas

concentrations with socioeconomic developments in a consistent way. Examples of these developments are population growth, climate change mitigation strategies, and economic relations between countries. The SSPs 1-2.6 and 5-8.5 are of special interest because they provide pathways for either reaching the 2-degree target or living in a world that remains reliant on fossil resources. Using these pre-defined scenarios enables us to capture the broadest range of possible feedbacks between climate change and alkalinity.

Temperature, as shown by our model, has the potential to greatly change alkalinity generation globally. We upscaled our model for normalized alkalinity concentration to the mid-latitudes (corresponding to the temperature range of 0.0–20.0 °C) to study the effect of increasing MAT on alkalinity flux (alkalinity concentration multiplied by mean annual river discharge). Our results show that the alkalinity flux decreases by up to 68% in the current MAT range of 15.0–17.5 °C. With MATs increasing, on average, by 1.4 and 3.6 °C within this temperature band until the end of this century in the low and high-emissions scenarios, respectively, normalized alkalinity concentration is projected to be shifted towards its minimum at -22.5 °C (refer to the schematic evolution of normalized alkalinity concentration in Fig. 3c, d). This results in a reduction of alkalinity flux by 33 and 68%, respectively. We propose this decrease is due to future aridification, which is projected to be especially pronounced in the Mediterranean, southwestern South America and western North America[35]. A rise in MAT within the 12.5–15.0 °C band is also anticipated to cause a decrease in alkalinity flux for both emissions scenarios, since alkalinity generation will depart from its optimal climatic conditions (high water and acid availability). We attribute this effect to the concomitant decrease in the solubility of carbonates. Further, in the 17.5–20.0 °C band, under both emission scenarios, the alkalinity flux is expected to decrease as it is pushed toward its minimum value around 22.5 °C. While we expect a further decrease in alkalinity flux for the temperature band 10.0–12.5 °C under SSP5-8.5, a small increase is seen under SSP1-2.6. Given the small increase in MAT expected in the latter (1.0 °C), the majority of the catchments remains in the optimal weathering range. In contrast, MAT under the SSP5-8.5 scenario will increase by 3.0 °C, pushing these catchments outside of their optimal weathering conditions. Finally, our model predicts a reduction in alkalinity flux for the temperature range of 0.0–2.5 °C under both emissions scenarios. This could be explained by a reduction in glacial cover, and thus the erosive force, in these regions. However, we would also expect a significantly large surface area of the fine-grained matrix to be exposed in these scenarios, which should in contrast cause weathering to increase. This contrast may be attributed to the fact that our dataset currently contains no known Arctic erosion rates, meaning that our model input data for this temperature band are not from the Arctic, but rather from high-altitude catchments like the Himalayas (Fig. 1a). Accordingly, we recommend that focus on erosion rate and alkalinity measurement in high-latitude catchments worldwide is now needed to enhance understanding of the direct impact of the ongoing rapid deglaciation on carbonate weathering.

We expect the alkalinity flux in catchments with historical MATs in the range of 5.0–10.0 °C to increase with advancing climate warming under both emissions scenarios. We attribute this to a greater acid availability from soil respiration, which increases with climate warming[36]. In both scenarios, alkalinity generation is pushed from its local minimum at -5 °C towards the local maximum at -12.5 °C. Based on data from 60 large rivers, other authors determined the highest alkalinity flux in temperate, very humid regions to be 72.3 t(bicarbonate) $a^{-1}$ $km^{-2}$ (ref. [37]). We expect the highest absolute increase in alkalinity flux of 30.1 t(bicarbonate) $a^{-1}$ $km^{-2}$ for the historical temperature band 7.5–10.0 °C under scenario SSP5-8.5 (Fig. 3f) and attribute this to a shift towards a more temperate and humid climate with sufficient acid availability and favorable carbonate solubility. For the temperature band 2.5–5.0 °C, the change in alkalinity flux is dependent

on the emissions scenario: Under SSP1-2.6, the catchments are moved further to the minimum weathering zone and a reduction in alkalinity flux of 3% is expected. In contrast, the temperature increase under SSP5-8.5 is far greater and allows for a transition out of the minimum weathering zone, resulting in a gain in terrestrial alkalinity flux of 11%. This is supported by prior findings of increased bicarbonate flux due to climate warming for the period 1961–2004 in eight rivers in Iceland[38].

Our results show that carbonate weathering will respond to temperature changes as long as moisture availability is constant or sufficient. Weathering reacts more extremely to stronger rising temperatures under the high-emissions scenario compared to the low-emissions one (refer to the thick and thin arrows in Fig. 3d, c, respectively). Because the size of our dataset was limited by the number of consistent erosion rates available, we could not cover the full range in model parameters (very low and very high values) that occur at mid-latitudes. The catchments that showed extreme values that were not covered by our model training dataset were excluded from the calculations of alkalinity flux changes in mid-latitudes (see "Methods"). The resulting bias might lead to an overestimation of the flux changes. The exclusion of all catchments with a mean annual runoff <150 mm $a^{-1}$ probably also omits relatively water-scarce catchments, in which weathering is generally lower. In addition, the erosion rate of 100 mm $ka^{-1}$ assumed for all catchments is within the *efficient erosion rate regime*, which has the potential to produce high amounts of alkalinity. In the following, our calculations of absolute change in alkalinity flux in mid-latitudes thus represent a good first estimate, but should be viewed with some caution because the model training dataset represented the crucial, yet not the full range of mid-latitude catchments.

In both scenarios, we expect the greatest increase in alkalinity flux in Central Europe and Central Asia, following a latitudinal band at -45°N (predicted for the world's largest basins which are characterized by a historical MAT of 0.0–20.0 °C, illustrated in Fig. 4a, b). While the main part of the USA is projected to have no significant change in alkalinity flux under scenario SSP1-2.6, in SSP5-8.5 a lower flux is anticipated. A reduced alkalinity flux is mainly associated with regions north of 60°N and south of 30°N.

We show that the mid-latitude (0.0–20.0 °C) alkalinity flux would decrease, on average, by 1.6 t(bicarbonate) $a^{-1}$ $km^{-2}$ in SSP1-2.6 and increase, on average, by 0.5 t(bicarbonate) $a^{-1}$ $km^{-2}$ in SSP5-8.5. We here assume that the change in terrestrial alkalinity flux is solely due to a change in alkalinity concentration. However, we did not take into account the combined climate change-induced impacts on weathering and freshwater discharge, as currently we insufficiently understand how the product of discharge and concentration responds to climate change (see Supplementary Figs. 9 and 10). Considering a mean global bicarbonate flux to the ocean of 19.4 t $a^{-1}$ $km^{-2}$ (ref. [37]), our results imply a reduction of the global alkalinity flux of -8% under SSP1-2.6, or an increase of -3% under SSP5-8.5 due to the projected change in mid-latitudes.

If we attribute this decrease/increase in riverine alkalinity flux solely to carbonate weathering, we can expect an increase/drawdown of atmospheric $CO_2$ of the same magnitude for mid-latitudes until the end of the century (assuming $CO_2$ to be the dominant source of acidity and half of the bicarbonate equivalents originating from $CO_2$). Thus, under SSP1-2.6, reduced carbonate weathering in mid-latitudes due to climate warming leads to a reduction in $CO_2$ sequestration ($\emptyset\Delta = +0.3$ tC $a^{-1}$ $km^{-2}$). In contrast, we expect an increase in carbonate weathering under SSP5-8.5, resulting in an additional short-term $CO_2$ sink ($\emptyset\Delta = -0.1$ tC $a^{-1}$ $km^{-2}$). Globally, chemical weathering currently drives $CO_2$ consumption of -2 tC $a^{-1}$ $km^{-2}$ (ref. [8]). Our values for a projected change in $CO_2$ release and consumption due to climate warming in mid-latitudes would affect this rate by -15% and 5%, respectively. In terms of global anthropogenic emissions per year (34.9 $GtCO_2$ in 2021)[39], the change we predict in $CO_2$ uptake or release is small. Under SSP5-8.5, an increase in carbonate weathering offsets -0.05% of

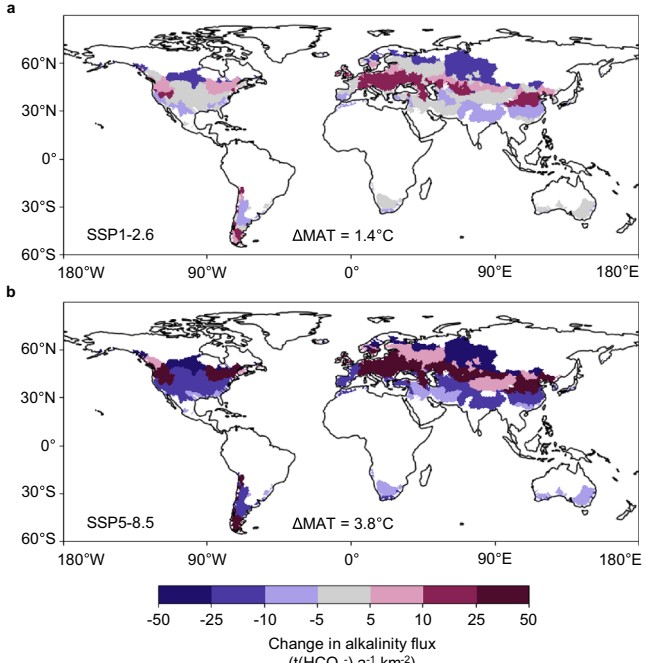

**Fig. 4 | Projected change in alkalinity flux due to climate warming.** Colors indicate the projected absolute change in alkalinity flux of catchments per temperature band for the historical temperature range of 0.0–20.0 °C, globally (corresponds to a land surface area of 44,506,993 km²), under scenarios **a** SSP1-2.6 and **b** SSP5-8.5. The mean change in mean annual temperature (ΔMAT) under SSP1-2.6 and SSP5-8.5 (SSP: shared socioeconomic pathway) until the year 2100 are projected to be 1.4 and 3.8 °C, respectively. Catchment areas in white were excluded from the analysis, since their historical MATs were lower or higher than the temperature range of 0.0–20.0 °C. For the calculation of the absolute alkalinity flux as specific mass flux, a molar mass of 61.02 g mol⁻¹ for bicarbonate (HCO₃⁻) was used, as at pH 7–9, the alkalinity concentration is approximately equal to the bicarbonate concentration[45,46]. The background maps with the continent outlines in (**a**) and (**b**) are from naturalearthdata.com[69].

anthropogenic emissions. We note that the change predicted by us refers only to the catchments with historical MAT of 0.0–20.0 °C, so that the magnitude of the change could be larger if all catchments globally were taken into account; especially considering that some of the alkalinity hot spots are located in areas with historical MAT > 20.0 °C. Moreover, it is a slow process that, as we show, is nevertheless influenced. This unfolds its effect over time and not via speed.

Our analysis of the global dataset on catchment alkalinity and erosion reveals that catchment average erosion rate is a first-order nonlinear control on alkalinity generation globally. Both the direction and degree of climate change-driven alteration of alkalinity fluxes depend on how strongly climate parameters will change in the future. To predict weathering rates at a global scale more precisely, we advocate for an expansion of ¹⁰Be erosion rate data where applicable, particularly in areas of under-represented areal carbonate proportion, temperature ranges and soil regolith thicknesses.

## Methods

### Water sampling and measurements

We collected 111 water samples for analysis of total alkalinity (AT) and dissolved inorganic carbon (DIC) at selected locations of ¹⁰Be erosion rate measurements during two sampling campaigns, one in Germany in May 2020 and another in Switzerland, Italy and Austria in June/July 2020. We collected the river water directly into 300-mL BOD bottles, added 300 μL of saturated mercury chloride solution and sealed the bottles with ground-glass stoppers, Apiezon type M grease and plastic caps (no-head space). The bottles were stored in the dark at ambient temperature. We used a Marianda VINDTA 3 C (Versatile Instrument for the Determination of Titration Alkalinity) to determine the DIC concentration by coulometric titration[40]. We analyzed the AT concentration by performing a potentiometric titration using a Metrohm 888 Titrando with an Aquatrode pH probe. We calibrated both instruments against certified reference materials (CRMs) provided by Andrew Dickson (Scripps Institution of Oceanography). We recorded in situ water temperature and electrical conductivity using a WTW Multi3430 with IDS TetraCon 925 and turbidity using a HACH 2100Qis. All measurements are reported at https://doi.pangaea.de/10.1594/PANGAEA.939660[41].

### Data requirements and sources

Since we wanted to build our analysis on a uniform set of ¹⁰Be erosion rates, we first extracted all locations of available ¹⁰Be erosion rate measurements from the OCTOPUS database[42]. (We note that there are other works not included in the OCTOPUS database.) We then assigned available alkalinity data in compliance with the following conditions: (i) The location of the alkalinity measurement should be in the same river as that of the erosion rate; ideally, the sampling locations of both measurements are identical. (ii) If the ideal condition of identical locations cannot be fulfilled, the location of the alkalinity measurement needs to be downstream of the erosion rate one so that the potential effects of an erosional event are captured in the alkalinity signal. (iii) The distance between the two locations should not exceed 25 current kilometers. (iv) One should try to exclude alkalinity locations that allow the inflow of tributaries larger than 10 current kilometers downstream of the erosion rate location. We made exceptions to these conditions for larger rivers (e.g., Maas or Neckar). We examined each location individually using QGIS 3.10.0.

For 111 of the 233 total erosion rate locations used in our analysis, we generated our own alkalinity data. We selected these sites because the respective catchments showed a large range in areal carbonate proportion, MAT and erosion rate, enhancing the diversity of our dataset. We ensured that there were no dams in the immediate vicinity of the sampling sites upstream (proportion of the catchment area affected by dams <50 %). In our dataset, 19 catchments are characterized by an areal proportion affected by dams of ≥10%. We performed a sensitivity test for normalized alkalinity concentration examining the influence of dams by removing all catchments characterized by an areal proportion affected by dams of ≥10% from the training dataset and running the GLM with the same set of covariates as in model M5. The resulting model function was similar to the original one of model M5 (comparison of both model functions is shown in Supplementary Fig. 11). We added 76 further alkalinity measurement locations from the GLORICH database[43], which met the above-defined requirements. We obtained the remaining 46 alkalinity values from individual published manuscripts or on request from government agencies. All original sources are listed at https://doi.pangaea.de/10.1594/PANGAEA.940522[44].

In addition to direct measurements of AT, we also accepted bicarbonate concentrations as alkalinity values. Alkalinity in rivers is approximately equal to the concentration of bicarbonate[45,46]. We were able to confirm this by calculating the concentrations of bicarbonate, carbonate and hydroxide using the program CO2SYS[47] for the samples from our sampling campaigns 2020. We used the measured values for AT, DIC, salinity, and water temperature together with the CO₂ constants from Millero[48]. The average proportion of bicarbonate in total alkalinity is 98 ± 4% (Supplementary Fig. 12).

Our dataset comprises erosion rates spanning four orders of magnitude (2–9829 mm ka⁻¹) and alkalinity covering a large range (4–4626 μmol L⁻¹). The alkalinity data include both single and time-series measurements (1–3940 measurements per location). For locations with multiple measurements (bi-monthly to monthly

measurements, sometimes decades-long studies), the mean was calculated, while locations with single measurements were regarded as mean annual values. The seasonal bias introduced by this simplification is tolerable, as, with a few exceptions, it is low compared to the entire range of alkalinity concentrations in our study (Supplementary Fig. 13). To circumvent the problem of concentration/dilution (see section "Some considerations on normalized alkalinity" below), we used normalized alkalinity concentration (alkalinity concentration divided by mean annual runoff) and excluded all observations with a mean annual runoff lower than 150 mm a$^{-1}$.

We combined the point sampling measurements of erosion rate and alkalinity concentration with the spatial description of runoff, lithology, temperature, precipitation, permanent snow and ice cover, forest cover, soil regolith thickness, and area affected by dams, of the respective catchment upstream from the erosion rate measurement location. We used the basin outlines from the OCTOPUS database[42] to compute the catchments' mean values using QGIS 3.10.0. We calculated mean annual runoff from the $0.5° \times 0.5°$ raster in the UNH/GRDC Composite Runoff Fields V1.0[49] to normalize the alkalinity measurements (alkalinity concentration divided by mean annual runoff). We determined the lithological coverage of the basins from the global lithological map database GLiM[50] by calculating the area of the individual rock classes as a percentage of the total catchment area. We combined two main classes ("sc" = "carbonate sedimentary rocks" with carbonate rocks being dominant; "sm" = "mixed sedimentary rocks" with carbonate being mentioned) and one subclass ("mtpu" = " metamorphics" with minor carbonate occurrences) indicative of carbonate presence into one class[50]. We computed the annual mean air temperature (MAT) and the annual mean precipitation (MAP), based on climate data for 1970–2000, from the $10' \times 10'$ raster of the WorldClim 2.1 database[51]. We extracted the areal proportion of permanent snow and ice as well as of forest cover from the $300\,m \times 300\,m$ raster of the GlobCover 2009 land cover map[52]. We took soil regolith thickness, defined as the depth from the surface to the bedrock, from the $1\,km \times 1\,km$ raster of the DTB (Global Depth to Bedrock) dataset[22]. We calculated the proportion of the watershed affected by dams using the global vector file from GOODD (GlObal geOreferenced Database of Dams)[53]. All data are summarized at https://doi.pangaea.de/10.1594/PANGAEA.940522[44].

## CO$_2$ consumption of carbonate and silicate weathering at different timescales
At pH 7–9, the alkalinity concentration is approximately equal to the bicarbonate ion concentration (-95% of the carbon in the water is in the form of bicarbonate ions), as the equilibrium between dissolved CO$_2$, bicarbonate and carbonate in this pH range is strongly in favor of bicarbonate[45,46]. While weathering of divalent cation silicates consumes two equivalents of CO$_2$, carbonate weathering consumes just one. Once in solution, the produced ions are transported via rivers to the ocean, where the bicarbonate ions can also dissociate into carbonate ions, depending on the buffer capacity of the ocean, i.e., oceanic alkalinity. In supersaturated waters with respect to calcium carbonate, such as at the ocean surface, the bicarbonate ions can react with calcium ions and about one equivalent of CO$_2$ is released into the atmosphere. The carbonate ions can be directly precipitated as calcium carbonate. Consequently, beyond the calcium carbonate compensation time (-10 ka)[2], only silicate weathering acts as a long-term sink for atmospheric CO$_2$ and carbonate weathering acts CO$_2$-neutral[1]. Terrestrial carbonate weathering initiated by volcanogenic or anthropogenic sulfuric acid instead of CO$_2$ also leads to CO$_2$ released to the atmosphere[20,54,55].

## Some considerations on normalized alkalinity
We had chosen to use the normalized alkalinity in order to be able to perform a dilution correction, appreciating the effects of surface runoff (intense rain events, snow melt), which we consider as alkalinity

free, as it is not impacted by weathering processes. If we had not done this, the alkalinity study would have been obscured by the influence of such surface runoff. Accordingly, we used the normalization as an analytical tool, the later quantitative integration does consider alkalinity concentration and actual runoff to establish alkalinity transports.

If we assume a scenario in which runoff increases over time, we see two possible cases: (1) Increased surface runoff leads to a dilution of the alkalinity concentration (pure rainwater or snow melt dilution effect) and accordingly also to a lowered normalized alkalinity. However, since we eventually multiply by runoff and by discharge (squaring the volume part), this dilution effect cancels out. The resulting alkalinity transport remains the same. (2) In this case, increased surface runoff causes a washout of groundwater containing alkalinity from reservoirs, so it does not cause a pure dilution of the alkalinity concentration, but may even cause a (temporary) increase in alkalinity concentration. Whether this will lead to a lower or higher normalized alkalinity, depends on the ratio of the changes in alkalinity and runoff, respectively. Independently of that ratio, the later integration (multiplication with the square of the runoff as described in the first case) will yield enhanced alkalinity transports, since in contrast to the first case, here in this case extra alkalinity has been added to the runoff.

We like to emphasize that the question whether alkalinity increases with runoff or not is one of the most crucial questions in Earth Sciences. However, we are not aware of any evidence in the literature indicating that an increase in runoff leads to a permanent increase in alkalinity. Rather, we think that after an alkalinity reservoir is washed out by a rain/snow melt event, the reservoir is depleted, resulting in a reduced concentration after the event. Thus, alkalinity generation is potentially not transport-limited, but the weathering itself, the chemical dissolution, is the limiting factor, i.e., kinetic control is exerted by chemical breakdown of the rock. This debate seems to be the "golden question" in the alkalinity issue.

## $^{10}$Be-derived erosion rates
Among other approaches, terrestrial erosion rates over scales of centuries to millennia are now frequently obtained from paleo-sediment flux measurements and terrestrial cosmogenic nuclides[16,42]. However, no approach is ideal. Conventional sediment-yield measurements can substantially underestimate long-term average erosion rates[56]. The analysis of cosmogenic nuclides, such as $^{10}$Be, in river sediment at the outlet of a catchment allows one to average total denudation rates over both the whole basin area and long time periods ($10^2–10^5$ a), also capturing infrequent high-magnitude events[15,16]. However, the $^{10}$Be approach requires that the landscape has achieved steady state with respect to the gradual removal of quartz, with predictable $^{10}$Be concentrations from quartz-bearing soils throughout a catchment (such as, e.g., a recently deglaciated catchment has not achieved). Where adequate appreciation for the assumptions and caveats of the sampling approach are considered, total erosion rates from cosmogenic nuclides can reproduce sediment-yield-derived erosion rates for the same catchment[57,58]. A significant caveat is that the $^{10}$Be-derived catchment erosion rate (a mass loss rate generally expressed in units of mm ka$^{-1}$) has a relatively long observation window (centuries to tens of thousands of years, for very fast and very slow eroding catchments), whereas active stream sediment discharge calculated for example from suspended load measurements are averaged over hour to decadal windows. Another caveat is that the $^{10}$Be-derived erosion rates consider the entire mass loss (chemical and detritus) whereas dissolved loads are not always measured for streams gauged for the suspended load. The best correspondence between $^{10}$Be and alkalinity would therefore be in a relatively flat stable catchment where chemical erosion dominates, and soil mass-depth is uniform over many millennia.

The averaging time scale of $^{10}$Be-derived erosion rates is a function of the erosion rate itself[16]. We calculated the averaging timescales for the erosion rates used in our study by dividing the

average cosmic ray absorption depth scale in regolith soils (~800 mm) by the erosion rate (Supplementary Fig. 14). While rapidly eroding catchments show the shortest averaging timescales of ~$10^2$ a, slowly eroding ones are characterized by averaging timescales of $10^4$–$10^5$ a. To test whether the long-term averaged [10]Be-derived erosion rates were related to the alkalinity signal, as proposed by our model, rather than a potential short-term disturbance in erosion rate, such as the construction of a dam, we included the proportion of the watershed affected by dams as a variable in the model. In our dataset, 20% of all catchments contained at least one dam. An areal proportion of the watershed affected by dams of at least 10% were found in 8% of all data points. Inclusion of the proportion of the watershed affected by dams in model M5 proved to be significant, but did not change the general trend in normalized alkalinity as a function of [10]Be-derived erosion rate and the other covariates of model M5 (Supplementary Fig. 15). Furthermore, another global study[59] determined short-term erosion rates (estimated from suspended sediment yield) to be only 1.4 times as high as long-term erosion rates (estimated from [10]Be concentrations) in temperate climates. Accordingly, we assume that the set of [10]Be-derived erosion rates used in our study represents the erosional behavior of a watershed well and that a potential temporal disjunction between [10]Be-derived erosion rate and alkalinity concentration has no decisive impact on the investigated relationships.

## Model preparation

We carried out the statistical analysis by using the packages "stats"[60] for generalized linear models (GLMs) and "mgcv"[61] for generalized additive models (GAMs) in R[60]. While determining the predictor variables for modeling normalized riverine alkalinity, we achieved the best model results by using a GAM.

We used GAM methodology to employ a fully data-driven approach to get a first impression of the general functional relationships of the independent variables to normalized alkalinity concentration. In our GAM specification, all covariates were included as smooth functions. We then fitted a GLM with polynomial functions of different degrees based on the findings of our GAM results, accepting a lower model fit of the GLM when compared to the GAM, for the following three reasons: (i) Some functions, which have been proven to demonstrate a linear relation (e.g., weathering and areal carbonate proportion)[9], showed "wiggliness" (lack of smoothness). Therefore, we restricted them by forcing a linear relation. The scientifically inexplicable "wiggliness" is most likely caused by our sample size; (ii) We wanted to ensure better comparability with the coefficients of previous studies[11,12] and thus used polynomial functions for the remaining variables; (iii) The leave-one-out cross-validation revealed a higher mean-squared prediction error for the GAM than for models M4 and M5 (Supplementary Table 1), suggesting that the GAM may overfit.

For the model and variable selection process, we used Akaike and Bayesian information criteria (AIC and BIC) and residual sum of squares (RSS). From this iterative process, we extracted five different models (M1–M5). While in the first model (M1) only the areal carbonate proportion is included as a predictor variable, our final model (M5) comprises five covariates. We used the GAM as reference (Supplementary Table 1).

Since normalized alkalinity concentration is a non-negative continuous response variable, we used a natural logarithm as the link function[62]. We tested the robustness of our model fit by performing a leave-one-out cross-validation, in which only a certain part of the data (number of observations−1) is set to fit the model, while the remaining part (1 observation) is used to test the model.

## Global riverine alkalinity function

As expected and published elsewhere[8–12], we identified lithology as the dominant control on alkalinity production (M1, Supplementary Table 1). We decided to use carbonate (sum of "sc", "sm", and "mtpu")[50] as the only lithological predictor variable for normalized alkalinity. The remaining rock types were not significant predictors (as revealed when considered in model M5, at a significance level of 0.05), with the exception of "intermediate plutonic rocks" ($P$ value = 0.007). As the latter lithology only covers 0.4% of the terrestrial Earth[50] and its inclusion improved the model score only minimally, we only incorporated carbonate into our models.

The discrepancy between observations and fitted model values was substantially reduced by adding MAT as a covariate (M2). To model the effect of MAT, we used a third (M2–M4) and a fifth (M5) degree polynomial function. To test whether the fifth-degree polynomial representation of MAT (M5) describes normalized alkalinity concentration better than the third-degree one (M3 and M4) in different climatic zones presented in our dataset, we divided our observations into temperature bands (5 °C steps) and determined the individual RSS. For all temperature bands, model M5 shows lower RSS than models M3 and M4 (Supplementary Table 2). Although the leave-one-out cross-validation for model M4 yielded an overall lower mean-squared prediction error (Supplementary Table 1), we chose M5 as our final model because the leave-one-out cross-validation revealed a lower or similar mean-squared prediction error for all individual temperature bands examined, with the exception of the temperature band [15 °C, 20 °C]. Accordingly, M5 performed better than M4 at the outer parts of our temperature range (Supplementary Table 3).

Model performance was further improved by adding (the natural logarithm of) physical erosion rate to the model (M3), revealing at a global level that physical erosion rate is also a first-order control on riverine alkalinity. Modeled normalized alkalinity as a function of erosion rate increases steeply until the *efficient erosion rate regime* is reached and then smoothly decreases again (Supplementary Fig. 2a).

We recognized that (the natural logarithm of) the catchment area, calculated as 2D area, and soil regolith thickness (depth to bedrock)[22], are two additional important predictor variables for normalized alkalinity concentration and these are considered in model M4 and our final model, M5.

Normalized alkalinity concentration in model M5 is calculated with the following equation:

$$
\begin{aligned}
\text{Normalized alkalinity concentration} = \exp[ & -1.163 \\
& + 0.01867(\text{areal carbonate proportion}) \\
& - 0.1504(\text{MAT}) \\
& - 0.009028(\text{MAT})^2 \\
& + 0.005944(\text{MAT})^3 \\
& - 0.0004681(\text{MAT})^4 \\
& + 0.00001007(\text{MAT})^5 \\
& + 0.2873(\ln(\text{erosion rate})) \\
& - 0.05615(\ln(\text{erosionrate}))^2 \\
& + 0.1342(\ln(\text{catchment area})) \\
& + 0.05078(\text{soil regolith thickness})]
\end{aligned}
$$

We did not include runoff as a predictor variable for normalized alkalinity, since we defined normalized alkalinity as alkalinity concentration per unit of runoff. The inclusion of runoff in the model would result in auto-correlation, which we need to avoid. We also examined other variables, such as areal proportion of snow and ice cover, vegetation (as areal proportion of forest cover), and MAP for their effects on normalized alkalinity. The first two were not significant when added to our final model M5. Precipitation was significant, but we decided not to include it into our final model because it only slightly improved the model score, and there was a high degree of multicollinearity (MAP and MAT showed a high correlation), which we wanted to avoid.

We also developed a GAM for alkalinity flux (alkalinity concentration × runoff) to test whether all covariates included in our

model M5 for normalized alkalinity are also significant predictors for alkalinity flux. We found that all five covariates (included as smooth functions) were significant predictors at a significance level of 0.001. The permutation feature importance test revealed a similar feature importance (areal carbonate proportion > catchment area > MAT > erosion rate > soil regolith thickness) as for our model M5 for normalized alkalinity (areal carbonate proportion > catchment area > erosion rate > MAT > soil regolith thickness).

The model results with all covariates and coefficients are shown in the R script (see "Data availability").

## Calculation of change in alkalinity flux due to climate warming

We calculated the terrestrial alkalinity flux on a catchment basis for catchments with historical (1980–2009) MAT of 0.0–20.0 °C. Global temperature fields for historical and future (2070–2099) periods were provided by ISIMIP[63], based on scenario simulations conducted with the GFDL-ESM4 for the CMIP6 project[64]. Together with atmospheric fields from the same source (precipitation, radiation, humidity and wind), we were able to run the global hydrological model HydroPy[65] to generate runoff and river discharge data. We compared historical with future data affected by climate warming according to a low (SSP1-2.6) and a high (SSP5-8.5) emissions scenario (SSP: shared socioeconomic pathway). In contrast to MAT, we kept the other covariates in model M5 constant. Since no global map for $^{10}$Be erosion rates exists, we assumed an erosion rate of 100 mm ka$^{-1}$ for all catchments. Further, we excluded all catchments whose values (in runoff, catchment area, soil regolith thickness) are not covered by our calibration data (i.e., runoff <150 mm a$^{-1}$, catchment area >239,000 km$^2$, soil regolith thickness <2.29 m and >22.85 m). Even though the largest 105 catchments are not included in the calculations due to this data filter, we applied an extrapolation to the entire land surface area for the temperature range of 0.0–20.0 °C (44,506,993 km$^2$), because the most important controlling factors (areal carbonate proportion, MAT and erosion rate) were not affected.

First, we fed our final model M5 with historical and future temperatures to yield estimates for historical and future normalized alkalinity, respectively. Second, we multiplied these with mean annual runoff (historical data: 1980–2009), which yielded alkalinity concentration. To assess only the effect of temperature change on alkalinity flux, we kept the discharge constant over time and multiplied both the historical and the future alkalinity concentration by the historical mean annual discharge.

## Data availability

The dataset used to develop the model in this study has been deposited in the PANGAEA database under the accession code https://doi.pangaea.de/10.1594/PANGAEA.940522[44]. The source data for the analysis of the effect of climate change on alkalinity flux generated in this study (data underlying Figs. 3 and 4) are provided in the Source Data file "Supplementary Dataset 1".

## Code availability

The R script, which includes all model results, can be accessed at https://github.com/nelelehmann/Alkalinity-responses-to-climate-warming-destabilise-the-Earth-s-thermostat/[66].

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

## Acknowledgements

We thank the German Federal Ministry of Education and Research (BMBF) for providing the funding for this project. N.L., H.T., and T.S. were supported by "The Ocean's Alkalinity: connecting geological and metabolic processes and time-scales", BMBF under "Make our Planet Great Again—German Research Initiative", grant number 57429828, implemented by the German Academic Exchange Service (DAAD). We thank M. Treblin, H. Treblin, P. Knobloch, L. Luitjens, and M. Ostermann for helping in sample collection. We thank P. Bartsch for measuring the total alkalinity concentration in the collected samples. We thank F.A.E. Roland and A.V. Borges for providing data on the Meuse basin. We thank L. Baldewein for helping with data stewardship. We thank the two reviewers for their thoughtful comments and efforts to improve our manuscript and the fruitful discussion that resulted.

## Author contributions

N.L., H.T., and J.G. designed the study; N.L. and S.L. participated in sample collection; N.L. measured the DIC concentration in the collected samples; J.H. contributed alkalinity measurements of the collected samples; N.L. and C.M. built the dataset; N.L. analyzed the data; N.L. and S.L. developed the model; T.S. provided the data for analyzing the effect of climate warming (from GFDL-ESM4 and HydroPy); N.L. wrote the draft with the help of H.T., H.L., J.G., and T.S., and with input from all other co-authors.

## Funding

## Competing interests

The authors declare no competing interests.
