## [Peer Review File · Nature Communications]

Alkalinity responses to climate warming destabilise the Earth's thermostatReviewer #1 (Remarks to the Author):

This paper deals with a very important and vexing problem for predicting future climate change and for our understanding of the carbon cycle: how will weathering rates respond to increasing pCO₂ and temperature in the future? The long-term response (how long our CO₂ "hangover" will last) depends on the answer. The work described seems like a new approach and reaches some conclusions that reinforce what has already been proposed, and proposes some new subtleties to appreciate. I support publication with a few suggestions below.

Abstract Line 19: It would be helpful to add the work "riverine" to the phrase "global alkalinity data."

Page 2 Line 8: The alkalinity generated by carbonate *and* silicate weathering (not just carbonate weathering) must be balanced by marine calcification, the only major sink for seawater alkalinity. If marine calcification only balanced carbonate weathering, then the alkalinity of the ocean would be increasing at a rate equal to silicate weathering, which does not appear to be the case.

Page 7 Line 20: I suspect "turbidity" is misused here - turbidity is the measure of clarity (vs opacity) of a liquid, whereas the passages before and after this mention deal with turbulence. Perhaps I am mistaken and turbidity is being used as a measure of how much erosion/sediment transport is going on? Because rivers carrying lots of suspended sediment are of course turbid. But either way, the awkwardly alternating use of turbidity and turbulence indicates that either the reader will be confused, or the authors are!

I recognize that the methods are described in that section after the main text, but I think the main text itself would benefit from just a couple of sentences describing what the authors actually did to go from observations to a model of the controls on weathering. This would go somewhere in the section "First-order controls on riverine alkalinity". As currently arranged, I was a bit baffled when the authors start to describe the findings without really stating what work was done and how.

Figure 3 is a bit hard to follow. First of all, why do panels c and d seemingly have more/different bins of MAT than panels a,b,e,f? And second, it took me a while to figure out where the purple line in c and d came from - is it the same as in figure 2a? If so that should be made clear.

Figure 4: These two panels show change in alkalinity flux for two scenarios, but at what point in those scenarios? Peak CO₂ / MAT? Maybe list on the figure the global MAT or pCO₂ for this "snapshot" of potential future climate.

Page 16 , line 22: It would be helpful to put the changes in CO₂ sequestration by carbonate weathering into the context of anthropogenic carbon emissions. Humans burned something like 10 GtC last year (that's probably out of date by now). How much CO₂ is sequestered each year by global weathering, and how much will that change in the various future scenarios based on your model? My intuition is that the change will be very small compared to anthropogenic forcing, and that should be noted. But this does not mean it is insignificant - it is very significant over the timescales at which weathering modulates the carbon cycle and climate (tens of thousands of years and longer).

One final remark: somewhere in the introduction the authors should explain the difference between carbonate and silicate weathering. They both generate alkalinity and sequester CO₂ on the short term. But on the long term, if carbonate weathering is balanced by marine carbonate burial (which removes alkalinity from seawater and increases [CO₂]), this leads to no net change in CO₂ (carbonate dissolution is the reverse reaction to carbonate precipitation). But, this is not true of silicate weathering - silicate weathering balanced by marine carbonate precipitation vs. burial does lead to

net removal of CO₂ from the atmosphere+ocean. So on long timescales it does matter how much of the weathering is from carbonates or silicate rocks.

Reviewer #2 (Remarks to the Author):

This study compiles 233 sampling locations over the global land surface and integrates the river chemistry with erosion rates (determined by ¹⁰Be), areal proportion of carbonate, mean annual temperature (MAT), catchment area and soil thickness to model the river alkalinity concentration using generalized linear models. As the study demonstrates, those factors are all first-order controls on river alkalinity. The author also couples the model with shared socio-economic pathways (SSPs) and shows that terrestrial alkalinity flux from mid-latitudes will be reduced and increased under the low-emission scenario and the high-emission scenario respectively.

While I find this study interesting and the thought is good in integrating the ¹⁰Be-derived erosion to other classical covariates to predict weathering flux on a global scale, I find a lot of issues that need to be addressed. At the current state, I don't think this is suitable to be published in NC.

Below are several of my major concerns.

1. The author uses runoff-normalized alkalinity (in the unit of mol a m⁻⁴) throughout this manuscript. This quantity is defined as the ratio of observed alkalinity in each river sample to the mean annual runoff of that river. Correct me if I am wrong, but I think this definition doesn't quite make sense. According to the author, this is used to mitigate the dilution issue. However, as far as I know, passing a certain threshold, at higher runoff values, the concentration of alkalinity (or bicarbonate) will get smaller. Hence, if a smaller concentration is divided by a higher runoff, the result (so-called runoff-normalized value) will become even smaller. I think this runoff-normalized value needs to be fully justified as it is the key to this paper.

2. Model performance and evaluation. The author uses GLM to predict the alkalinity concentration and tries to rank the relative importance of different parameters in regulating the alkalinity. A couple of questions here. A) The author gives AIC, BIC, and RSS. However, I also think adjusted R² is needed as it explains how much variance the model can explain. The current AIC, BIC, and RSS couldn't yield such information. B) The author mentions the leave-one-out cross-validation method, which is similar to AIC or BIC to reach a bias-variance balance, but the result is never mentioned or discussed. C) The author mentions first-order control multiple times throughout the manuscript, then what is the relative importance of each parameter in driving the change of the target variable? Do we know what is more important? Temperature, erosion, carbonate area, soil thickness or catchment area? D) As precipitation and temperature is highly correlated, I suggest the author uses precipitation instead of temperature in M5 and see what happens. If the result (AIC, BIC or adjusted R²) is better than that when using temperature, can we say precipitation is the dominant factor in regulating weathering flux, instead of temperature? P-value is also needed to test whether there is any significance between the predictor variable and target variable.

3. Water quality. From extended data fig. 9, a larger proportion of the data belongs to the category that has a single measurement, which will significantly bias the result. Normally, at least 1 data point in each season is needed to calculate the mean annual concentration. I don't think the current approach is suitable. In addition, how does the author obtain the annual mean alkalinity concentration? This can be tricky if not taken care of. It will be good to do the discharge-weighting by day, month and then year gradually to have a more robust discharge-weighted alkalinity concentration (which I personally think is better than the current runoff-normalized value)

4. The MAT effect. From fig. 2, we see an overall increase of alkalinity concentration

beyond 20 C. First, explanation is needed for the behavior. Second, from fig. 2b, a big proportion of land surface has MAT bigger than 17.5 (including the regions having a very high weathering rate, such as SE Asia) but is neglected in this study (outside of the 0-17.5 range). Hence, I would think the author should include this >17.5 region, which will also be useful in coupling with the SSPs.

5. The future prediction using SSPs. The author states that in their prediction, the runoff is held constant and the change in terrestrial alkalinity flux is entirely due to a change in alkalinity concentration. This is really not the case in reality. As runoff can be a big driver in changing the weathering flux, this should be included in the final model prediction when applied to SSPs.

Below are my minor comments:

The model's name (GLM) and its overall property and advantage should be mentioned much earlier. In the current manuscript, the author discusses a lot based on the model results before actually telling the reader what the model is (it only appears in the method part).

Line 5 page 4: what is "sediment discharge"?

Line 5 page 5: Why does the high erosion rate produce little alkalinity? kinetic-limited?

Line 10 page 5: The author states "Our global dataset indicates that areal carbonate proportion has a first-order positive effect on normalized alkalinity concentration". How does the author draw this conclusion? Any support from the regression outcome? How does it compare with temperature? I think a permutation test might be able to answer this question. This also applies to testing the importance of other parameters.

Line 20 page 5: how do the authors define "high amounts of alkalinity"?

Line 10 page 6: low temperature can also limit the reaction rate based on the Arrhenius equation. How does this temperature effect compare with the respiration effect?

Line 5-10 page 7: I don't see how erosion rate is related to the acid availability based on the author's data. The author needs to elaborate on this aspect as it is important for understanding the erosion control on CO₂ degassing. In addition, what is actually controlling the CO₂ degassing? Is it altitude, as suggested by reference 9 and 12, or erosion (aka, denudation), or slope? More discussions are needed here. In Line 20, p-value is needed to judge whether turbulent flow and erosion are related.

Line 5 in page 8: The explanation for the relationship between watershed area and normalized alkalinity concentration is poor. First, why can larger watersheds capture more precipitation? If we calculate precipitation per unit area, I don't think we can reach this conclusion. In addition, since the concentration is already normalized by runoff, this precipitation effect should be minimalized. Second, the author argues that "a larger watershed may have a larger relief", I don't think this is the case. Any study to support this? Also, the example in the following uses the "small, steep Japanese basins", which suggests smaller basins are steep and this is in direct conflict with what the author says?

In Line 10 page 8: I find it is hard to grasp what the author wants to convey here. In extended figure 2c, the normalized alkalinity concentration increases monotonically with soil thickness, which conflicts with the notion that a thicker soil will reduce the interaction between water and fresh mineral surfaces, thus reducing weathering rate. Also, thick soil will prevent soil production. I appreciate the author includes both soil thickness and erosion as covariates. Using multiple linear regression, the meaning of the coefficient for each variate is: when fixing the other parameters, how the target variable

will change given one unit change of that specific variate (as the author shows in extended data fig. 2). In this case, it means at constant erosion rate, when the soil thickness is increasing, how the weathering rate will change. I think the relationship between soil thickness and weathering requires more explanation from both a mechanistic perspective and the equation form.

Line 10 page 9: The author states "In our dataset, normalized alkalinity increases with the areal extent of permanent snow and ice cover (for all catchments with permanent snow and ice cover >1%)". Is this only for watersheds that have MAT < 2.5 C? An extended figure will be beneficial here.

Line 10 page 10: should "qualitative" be "quantitative" as the narrower temperature range is much more robust?

Line 10 page 10: The author states "Outside this range, (i) the implicit constancy of covariates (mean global carbonate extent = 22%³², erosion rate = 100 mm ka⁻¹) appears unrealistic for these regions". Does it mean the <0 C and > 17.5 C watersheds have constant carbonate extent and erosion rate? Please elaborate.

Extended Data Fig. 4: No need to use adjusted R² since there is only 1 predictor variable

Extended Data Fig. 4 and 5, p-values are needed.

[-3,0[and the other ranges. I find the "[" kind of not the normal form. Does this mean excluding 0 here?

Line 15 page 23. Dam <50% is still a significant proportion. I would suggest the author does some sensitivity tests (such as Dam < 10%). Line 23, I think salinity is also needed to calculate the bicarbonate concentration. Could the author give the salinity data?

Line 15 page 25, bicarbonate reacts with Ca to release CO₂. Carbonate ion will react with Ca and directly precipitate without releasing CO₂

Line 5 page 26: The author states "Where adequate appreciation for the assumptions and caveats of the sampling approach is considered, total erosion rates from cosmogenic nuclides often reproduce sediment-yield derived erosion rates for the same catchment". But the author also argues that sediment-yield measurements can substantially underestimate erosion rates. So ¹⁰Be will also suffer this drawback?

Line 5 page 28: The author states the other rock types are tested by M5 but there is no description in M5 that different rock types are tested.

Authors' Response to Reviewers' Comments**Reviewer #1**

#	Reviewer's Comment	Authors' Response	Position in Manuscript (Simple Mark-up)
1	It would be helpful to add the work “riverine” to the phrase “global alkalinity data.”	We agree. We added it.	p. 1, l. 19
2	The alkalinity generated by carbonate *and* silicate weathering (not just carbonate weathering) must be balanced by marine calcification, the only major sink for seawater alkalinity. If marine calcification only balanced carbonate weathering, then the alkalinity of the ocean would be increasing at a rate equal to silicate weathering, which does not appear to be the case.	We agree. We added it. We had not initially included silicate weathering, as our results below focus exclusively on carbonate weathering.	p. 2, l. 8
3	I suspect “turbidity” is misused here - turbidity is the measure of clarity (vs opacity) of a liquid, whereas the passages before and after this mention deal with turbulence. Perhaps I am mistaken and turbidity is being used as a measure of how much erosion/sediment transport is going on? Because rivers carrying lots of suspended sediment are of course turbid. But either way, the awkwardly alternating use of turbidity and turbulence indicates that either the reader will be confused, or the authors are!	We deleted the sentence about turbidity because we understand your objection that there could be confusion with turbulence. We originally thought that we could use turbidity as an indicator of secondary precipitation of calcium carbonate in the rivers. However, most of the turbidity is most likely due to suspended sediment (rather than secondary precipitation). Moreover, the correlation between erosion rate and turbidity was low anyway.	p. 7, l. 23 ff.
4	I recognize that the methods are described in that section after the main text, but I think the main text itself would benefit from just a couple of sentences describing what the authors actually did to go from observations to a model of the controls on weathering. This would go somewhere in the section “First-order controls on riverine alkalinity”. As currently arranged, I was a bit baffled when the authors start to describe the findings without really stating what work was done and how.	We added four sentences in the beginning of the section “First-order controls on riverine alkalinity” introducing the general structure of our approach in this paper (going from the observations in our data set to the model results) and presenting the main benefits of the model we used.	p. 3, l. 15 ff.

5	Figure 3 is a bit hard to follow. First of all, why do panels c and d seemingly have more/different bins of MAT than panels a,b,e,f? And second, it took me a while to figure out where the purple line in c and d came from - is it the same as in figure 2a? If so that should be made clear.	Yes, we agree. We have adjusted the figure so that the MAT bins are now all aligned. We added another temperature bin [17.5, 20.0 °C] to all panels, as we expanded our temperature range to 20.0 °C. Also, we mentioned in the caption that the purple line in c and d is the same as in Fig. 2a.	p. 14, l. 9
6	Figure 4: These two panels show change in alkalinity flux for two scenarios, but at what point in those scenarios? Peak CO₂ / MAT? Maybe list on the figure the global MAT or pCO₂ for this “snapshot” of potential future climate.	The two panels describe the change in alkalinity flux according to the different SSPs until the end of the century (2100). We added a sentence to clarify this in the figure caption and added a short information about the projected mean change in MAT (ΔMAT) according to the different SSPs on the figure.	p. 17, l. 5 ff.
7	It would be helpful to put the changes in CO₂ sequestration by carbonate weathering into the context of anthropogenic carbon emissions. Humans burned something like 10 GtC last year (that's probably out of date by now). How much CO₂ is sequestered each year by global weathering, and how much will that change in the various future scenarios based on your model? My intuition is that the change will be very small compared to anthropogenic forcing, and that should be noted. But this does not mean it is insignificant - it is very significant over the timescales at which weathering modulates the carbon cycle and climate (tens of thousands of years and longer).	Good point. The change is indeed very small compared to anthropogenic forcing (~0.05%). We added three sentences on this aspect at the end of the paragraph.	p. 18, l. 13 ff.
8	One final remark: somewhere in the introduction the authors should explain the difference between carbonate and silicate weathering. They both generate alkalinity and sequester CO₂ on the short term. But on the long term, if carbonate weathering is balanced by marine carbonate burial (which removes alkalinity from seawater and increases [CO₂]), this leads to no net change in CO₂ (carbonate dissolution is the reverse reaction to carbonate precipitation). But, this is not true of silicate weathering - silicate weathering balanced by marine carbonate precipitation vs. burial does lead to net removal of CO₂ from the atmosphere+ocean. So on long timescales it does matter how much of the weathering is from carbonates or silicate rocks.	We already had a section on this topic in the Methods (“CO₂ consumption of carbonate and silicate weathering at different time scales”). Nevertheless, we have now also added a sentence at the end of the first paragraph of the introduction.	p. 2, l. 10 ff.

Reviewer #2

#	Reviewer's Comment	Authors' Reponse	Position in Manuscript
1	The author uses runoff-normalized alkalinity (in the unit of mol a m-4) throughout this manuscript. This quantity is defined as the ratio of observed alkalinity in each river sample to the mean annual runoff of that river. Correct me if I am wrong, but I think this definition doesn't quite make sense. According to the author, this is used to mitigate the dilution issue. However, as far as I know, passing a certain threshold, at higher runoff values, the concentration of alkalinity (or bicarbonate) will get smaller. Hence, if a smaller concentration is divided by a higher runoff, the result (so-called runoff-normalized value) will become even smaller. I think this runoff-normalized value needs to be fully justified as it is the key to this paper.	We wanted to characterize alkalinity in a volume-independent manner and therefore considered the alkalinity concentration per unit runoff. Later, to study the change in alkalinity flux due to climate change, we considered runoff again. We therefore believe that the use of the runoff-normalized values is justified.	p. 3, l. 16 ff.
2a	Model performance and evaluation. The author gives AIC, BIC, and RSS. However, I also think adjusted R2 is needed as it explains how much variance the model can explain. The current AIC, BIC, and RSS couldn't yield such information.	We added the adjusted R² for each model in Extended Data Table 1.	p. 54
2b	Model performance and evaluation. The author mentions the leave-one-out cross-validation method, which is similar to AIC or BIC to reach a bias-variance balance, but the result is never mentioned or discussed.	Yes, that's true. We added a sentence in the Methods section under "Model preparation", mentioning that the "leave-one-out cross-validation revealed a higher mean squared prediction error for the GAM than models M4 and M5, suggesting that the GAM may overfit." Further, we added two sentences in the Methods section under "Global riverine alkalinity function", saying that "Although the leave-one-out cross-validation for model M4 yielded an overall lower mean squared prediction error, we chose M5 as our final model because the leave-one-out cross-validation revealed a lower or similar mean squared prediction error for all individual temperature bands examined, with the exception of the	p. 30, l. 25 ff. / p. 32, l. 3 ff.

	Model performance and evaluation. The author mentions first-order control multiple times throughout the manuscript, then what is the relative importance of each parameter in driving the change of the target variable? Do we know what is more important? Temperature, erosion, carbonate area, soil thickness or catchment area?	temperature band [15°C, 20°C]. Accordingly, M5 performed better than M4 at the outer parts of our temperature range.” We ordered the covariates in the models (in M1 – M5 in Extended Data Table 1) according to their importance. We demonstrated this by performing an AIC Stepwise Algorithm: If only one dependent variable could be added to the model to obtain the best AIC; it would be areal carbonate proportion. If another variable could be added, it would be MAT. The third, fourth and fifth variables would be erosion rate, catchment area, and soil thickness, respectively. → refer to “Declare importance of each dependent variable by AIC Stepwise Algorithm” in the provided R-script	R-script
2d	Model performance and evaluation. As precipitation and temperature is highly correlated, I suggest the author uses precipitation instead of temperature in M5 and see what happens. If the result (AIC, BIC or adjusted R2) is better than that when using temperature, can we say precipitation is the dominant factor in regulating weathering flux, instead of temperature? P-value is also needed to test whether there is any significance between the predictor variable and target variable.	We tested for this and the results when adding precipitation instead of MAT to model M5 are worse than for the “original” model M5 with MAT. → refer to model M6 in the provided R-script The p-values for precipitation in model M6 can also be found in the provided R-script. At this point, to avoid any confusion, we would like to point out that when the coefficients of the variables are compared, attention should be paid to whether the polynomials are orthogonalized or specified as raw polynomials.	R-script
3	Water quality. From extended data fig. 9, a larger proportion of the data belongs to the category that has a single measurement, which will significantly bias the result. Normally, at least 1 data point in each season is needed to calculate the mean annual concentration. I don't think the current approach is suitable. In addition, how does the author obtain the annual mean alkalinity concentration? This can be tricky if not taken care of. It will be good to do the discharge-weighting by day, month and then year gradually to have a more robust discharge-weighted alkalinity concentration (which I personally think is better than the current runoff-normalized value).	We cannot dismiss this problem, but we cannot resolve it with the data we have. One of the goals of our study was to investigate the influence of erosion rate on alkalinity generation, in addition to well-studied influencing factors such as lithology. Previous studies have often used other variables such as watershed slope as a proxy for erosion rate. However, we wanted to use real measurements. Accordingly, we chose to use ¹⁰Be-derived catchment-wide average erosion rates. There are not many such measurements available worldwide, as the method is still quite new and costly. Accordingly, this was the limiting factor. We investigated where globally these	p. 27, l. 2 ff

		measurements were made and tried to assign alkalinity measurements (see applied criteria in the Methods section). In order to obtain an appropriately large sample size, we took and measured alkalinity samples ourselves at known ¹⁰Be measurement sites in Europe. Due to cost and time constraints, we were only able to make single measurements. If we removed all single measurements, our dataset would be reduced from 233 observations to only 74. We know that seasonal biases can potentially distort the results, but we think that with our approach we have a good understanding of how erosion rate (and MAT, areal carbonate proportion, catchment area and soil thickness) influences alkalinity generation.	
4	The MAT effect. From fig. 2, we see an overall increase of alkalinity concentration beyond 20 C. First, explanation is needed for the behavior. Second, from fig. 2b, a big proportion of land surface has MAT bigger than 17.5 (including the regions having a very high weathering rate, such as SE Asia) but is neglected in this study (outside of the 0-17.5 range). Hence, I would think the author should include this >17.5 region, which will also be useful in coupling with the SSPs.	We added a sentence at the end of the paragraph. We decided to include only one more temperature bin (17.5-20.0°C), as our data set beyond this temperature (>20.0°C) does not cover the other co-variables well. See newly added sentences in the text about out-of-sample prediction regarding areal carbonate proportion for catchments with MAT >20.0°C. Due to this additional assessment, the overall mean change in alkalinity flux under the two different emission scenarios were also changed. We adjusted the text and figures accordingly.	p. 10, l. 11 ff. / p. 11, l. 13 / p. 13, l. 12 ff.
5	The future prediction using SSPs. The author states that in their prediction, the runoff is held constant and the change in terrestrial alkalinity flux is entirely due to a change in alkalinity concentration. This is really not the case in reality. As runoff can be a big driver in changing the weathering flux, this should be included in the final model prediction when applied to SSPs.	We added two extended figures where the change due to a change in temperature (i.e., alkalinity concentration) and discharge is considered. Refer to Extended Data Fig. 9 and 10. Here, the impact of the combined changes of alkalinity concentration and river discharge on alkalinity runoff (i.e., the product of alkalinity concentration and river discharge) is shown. In contrast, Fig. 3 and 4 show the influence of only the alkalinity concentration on the alkalinity flux (discharge is kept constant), since we cannot resolve the influences of the two individual parameters in a combined signal. However, this	p. 17, l. 15 ff. / p. 46-48

		very attribution (changes in alkalinity) is the main focus of our paper.	
6	The model's name (GLM) and its overall property and advantage should be mentioned much earlier. In the current manuscript, the author discusses a lot based on the model results before actually telling the reader what the model is (it only appears in the method part).	We added four sentences in the beginning of the section "First-order controls on riverine alkalinity" introducing the general structure of our approach in this paper (going from the observations in our data set to the model results) and presenting the main benefits of the model we used.	p. 3, l. 18 ff.
7	What is "sediment discharge"?	We substitute "stream dissolved load" to be more precise.	p. 5, l. 1
8	Why does the high erosion rate produce little alkalinity? kinetic-limited?	We believe that acid availability is the limiting factor in rapidly eroding catchments (see explanations on page 7 "Beyond peak alkalinity, at intermediate to high erosion rates (>100 mm ka ⁻¹), ..."), so alkalinity production in these catchment is equilibrium-limited. We added a sentence mentioning this on page 5, and referred to page 7 for further explanation.	p. 5, l. 2 ff.
9	The author states "Our global dataset indicates that areal carbonate proportion has a first-order positive effect on normalized alkalinity concentration". How does the author draw this conclusion? Any support from the regression outcome? How does it compare with temperature? I think a permutation test might be able to answer this question. This also applies to testing the importance of other parameters.	Good idea! We performed a permutation test. It revealed that areal carbonate proportion is the most important covariate in M5. → refer to "Declare importance by permutation feature importance test" in the provided R-script	R-script
10	How do the authors define "high amounts of alkalinity"?	As the third quartile of normalized alkalinity concentration in our dataset is ~2.1 mol a m ⁻⁴ , we define high amounts of alkalinity being above 2.1 mol a m ⁻⁴ , corresponding to an alkalinity concentration of ~2010 μmol L ⁻¹ . We found that for catchments with an areal carbonate proportion of less than 50%, an erosion rate within the efficient erosion rate regime and a MAT of ~10 °C (see the green line in Extended Data Fig. 1), normalized alkalinity is predicted by our model M5 to be ~2.5-5.0 mol a m ⁻⁴ , thus showing high amounts of alkalinity. We have added the values of the third quartiles of the normalized alkalinity concentration and the alkalinity	p. 5, l. 25 ff.

		concentration in parentheses at the end of the sentence to help the reader contextualise our results.	
11	Low temperature can also limit the reaction rate based on the Arrhenius equation. How does this temperature effect compare with the respiration effect?	Good point. Both decreased reaction rates according to the Arrhenius equation and decreased acid availability due to reduced respiration have the same negative effect on alkalinity production. We have included a corresponding sentence mentioning the temperature effect of the Arrhenius equation.	p. 6, l. 18 ff.
12	I don't see how erosion rate is related to the acid availability based on the author's data. The author needs to elaborate on this aspect as it is important for understanding the erosion control on CO2 degassing. In addition, what is actually controlling the CO2 degassing? Is it altitude, as suggested by reference 9 and 12, or erosion (aka, denudation), or slope? More discussions are needed here. In Line 20, p-value is needed to judge whether turbulent flow and erosion are related.	Since altitude and slope are closely related to erosion rate (which is also evident from our data set), it is difficult to single out a control factor responsible for CO₂ degassing. We think that in our model erosion rate is representative of the two other parameters (altitude and slope). To highlight close connectivity of all three parameters, we have added three sentences at the end of the paragraph. Moreover, we deleted the sentence about turbidity because we understand the objection of the other reviewer that there could be confusion with turbulence. We originally thought that we could use turbidity as an indicator of secondary precipitation of calcium carbonate in the rivers. However, most of the turbidity is most likely due to suspended sediment (rather than secondary precipitation). Moreover, the correlation between erosion rate and turbidity was low anyway.	p. 7, l. 23 ff.
13	The explanation for the relationship between watershed area and normalized alkalinity concentration is poor. First, why can larger watersheds capture more precipitation? If we calculate precipitation per unit area, I don't think we can reach this conclusion. In addition, since the concentration is already normalized by runoff, this precipitation effect should be minimalized. Second, the author argues that "a larger watershed may have a larger relief", I don't think this is the case. Any study to support this? Also, the example in the following	We agree with most of what this reviewer is saying. We'd like to suggest an alternative interpretation...but better over a phone/video conversation. But for now..., a simple response is that the stream power law for sediment discharge is as follows: Erosion = kA^mS^n, where A is catchment area, which is a simple but widely applied and useful proxy for stream discharge, which is a significant control on average erosion rate for most catchments. The S is the mean slope of the valley walls (i.e.	p. 8, l. 11 ff.

	uses the “small, steep Japanese basins”, which suggests smaller basins are steep and this is in direct conflict with what the author says?	not the stream gradient), the steeper slopes are going to provide more erosion power. K, m, and n are just constants and exponents that are fit to given areas around the planet. So, if we were just thinking about precipitation on a single pixel, and averaging those for an entire catchment of pixels, the reviewer would be right. But erosion, even slope wash or stream incision, all depend on the size, because toward the bottom of the slope there will be more water and more erosion power, and in a larger order stream there will be more erosion power, so for a larger A and S there is more erosion, on average. There are particular exceptions, but that is true for every rule...and this is a rule, not a law. We don't know of any empirical equation that has been used to link catchment size and relief.	
14	I find it is hard to grasp what the author wants to convey here. In extended figure 2c, the normalized alkalinity concentration increases monotonically with soil thickness, which conflicts with the notion that a thicker soil will reduce the interaction between water and fresh mineral surfaces, thus reducing weathering rate. Also, thick soil will prevent soil production. I appreciate the author includes both soil thickness and erosion as covariates. Using multiple linear regression, the meaning of the coefficient for each variate is: when fixing the other parameters, how the target variable will change given one unit change of that specific variate (as the author shows in extended data fig. 2). In this case, it means at constant erosion rate, when the soil thickness is increasing, how the weathering rate will change. I think the relationship between soil thickness and weathering requires more explanation from both a mechanistic perspective and the equation form.	Yes, this is right. We would also have expected normalized alkalinity concentration to decrease above a certain soil thickness. However, this is not evident from our data set. To support this, we have included a figure showing normalized alkalinity concentration as a function of soil thickness and erosion rate as Extended Data Fig. 5. Only for the four data points with the greatest soil thickness (soil thickness >20 m), we can observe low normalized alkalinity concentrations. However, the general trend evident from our data is a steady increase with increasing soil thickness ($R^2 = 0.1$). If more data points that have higher soil thickness (>25 m) were added to our data set, a decreasing trend might also be observed. This aspect should be investigated in future studies with a larger data set covering higher soil thicknesses. We rewrote the end of the section to remove the confusion of high normalized alkalinity concentration for high soil thicknesses, referred to Extended Data Fig. 5, and deleted the part where we used erosion rate as an explanation for the observed trend.	p. 9, l. 7 ff. / p. 42

		As we used the “depth to bedrock” database to calculate soil thickness, maybe it would be more accurate to use “regolith thickness” instead of “soil thickness” in our manuscript to avoid confusion about the high values (up to 20 m)? We added this figure as Extended Data Figure 6.	p. 10, l. 3 / p. 43								
15	The author states “In our dataset, normalized alkalinity increases with the areal extent of permanent snow and ice cover (for all catchments with permanent snow and ice cover >1%)”. Is this only for watersheds that have MAT < 2.5 C? An extended figure will be beneficial here.										
16	Should “qualitative” be “quantitative” as the narrower temperature range is much more robust?	Yes, good point! We changed it.	p. 11, l. 11								
17	The author states “Outside this range, (i) the implicit constancy of covariates (mean global carbonate extent = 22% ³² , erosion rate = 100 mm ka ⁻¹) appears unrealistic for these regions”. Does it mean the <0 C and > 17.5 C watersheds have constant carbonate extent and erosion rate? Please elaborate.	We understand how this could be confusing. For the later global assessment of the change in alkalinity flux due to climate warming, we fed our model with calculated values for all catchments included in the global analysis (areal carbonate content, catchment area, soil thickness and MAT). Since no global map for ¹⁰ Be erosion rates exists, we assumed a constant erosion rate of 100 mm ka ⁻¹ for all catchments. Our dataset showed that this rather low erosion rate does not fit to catchments with MAT <0°C. We did not explain the out-of-sample prediction problem regarding areal carbonate proportion for catchments with MAT >20°C very well here. We revised the sentence and added a few explanatory sentences.	p. 11, l. 13 ff.								
18	Extended Data Fig. 4: No need to use adjusted R2 since there is only 1 predictor variable	Okay, we used R ² instead.	e.g. p. 7, l. 14								
19	Extended Data Fig. 4 and 5, p-values are needed.	Yes, we added them.	p. 41, l. 3								
20	[-3,0[and the other ranges. I find the “[” kind of not the normal form. Does this mean excluding 0 here?	Yes, we used “[”, so the 0 does not show up in two bins, but only in the next bin [0,5[. We used the notation according to ISO 31-11:   [ [a,b] right half-open interval in \mathbb{R} from a (included) to b (excluded) [a,b] = {x ∈ ℝ a ≤ x < b}   ( (a,b)     We could change our notation from two square brackets to one square brackets and one parenthesis, if that is preferred?	[[a,b]	right half-open interval in \mathbb{R} from a (included) to b (excluded)	[a,b] = {x ∈ ℝ a ≤ x < b}	((a,b)			p. 55
[[a,b]	right half-open interval in \mathbb{R} from a (included) to b (excluded)	[a,b] = {x ∈ ℝ a ≤ x < b}								
((a,b)										

21	Dam <50% is still a significant proportion. I would suggest the author does some sensitivity tests (such as Dam < 10%). Line 23, I think salinity is also needed to calculate the bicarbonate concentration. Could the author give the salinity data?	We performed a sensitivity test for dam extent <10%. The model results were similar to the ones of the original model M5 (see added Extended Fig. 11). Yes, salinity is also needed for the calculation. We added that. The salinity data is available at https://doi.pangaea.de/10.1594/PANGAEA.939660, which is mentioned in the Methods section under “Water sampling and measurements”.	p. 26, l. 9 ff. / p. 26, l. 24 / p. 50
22	Bicarbonate reacts with Ca to release CO₂. Carbonate ion will react with Ca and directly precipitate without releasing CO₂	Yes, we adapted the text accordingly.	p. 28, l. 13 ff.
23	The author states “Where adequate appreciation for the assumptions and caveats of the sampling approach is considered, total erosion rates from cosmogenic nuclides often reproduce sediment-yield derived erosion rates for the same catchment”. But the author also argues that sediment-yield measurements can substantially underestimate erosion rates. So ¹⁰Be will also suffer this drawback?	We are not sure that there are data to support that sediment yield can reproduce ¹⁰Be erosion rates. It depends on how well the bedload, suspended load, and dissolved loads were all determined. The ¹⁰Be erosion rates are ‘all mass’, averaged over millenia, but few streams actually have all three loads measured even over a decade. Most streams just have suspended loads. If only suspended load is reported and converted to erosion rate, then the ¹⁰Be-derived erosion rate should be higher. If an appropriate empirical model is applied to convert suspended load to total load, then the erosion rate derived from this may be more equivalent to ¹⁰Be-derived erosion rates, BUT the same issue applies, that the ¹⁰Be is observing over millennia, whereas measured loads are rarely measured over windows greater than a century. So, while there are many drawbacks for ¹⁰Be erosion rate interpretations, the issue is that if surface processes are operating at different rates over different timescales, then the two approaches may not yield equivalent results. However, in (even tectonically active) catchments where sediment flux is relatively constant over the past few millennia (this timescale will be shorter if the rates are very high), it has been shown that ¹⁰Be-derived erosion rates compare to not just sediment	p. 29, l. 6 ff.

		flux-derived erosion rates, but even to exhumation rates derived from low-temperature thermochronology.	
24	The author states the other rock types are tested by M5 but there is no description in M5 that different rock types are tested.	We added the information about the significance-level in the text. All p-values and adjusted R ² for model M5, when a different rock type is added, can be found in the provided R-script. → refer to “pv_rock_type” and “adj_R_sq_rock_type” in the provided R-script	p. 31, l. 18

Reviewer #2 (Remarks to the Author):

I appreciate the author's efforts in addressing my concerns and making the text more polished. I think the current text is more accessible to the readers. However, I feel some of my major concerns are still not resolved by the author. I feel those concerns are critical for this paper to make a big impact in carbon cycle community and I hope these concerns can be further addressed and discussed to some extent.

In several responses, the author told the reviewer to see the provided R-script. But I am sorry that I didn't find any R-script in the uploaded files.

1) I am still not convinced about using the normalized alkalinity concentration to represent the weathering behavior. Here is one thought experiment. Imagine a single watershed with a time-series set of data. Assuming the runoff is changing from small to big over time. If the weathering can keep up with the runoff (for example, we see a linear relationship between the absolute amount of elements released from rocks and the runoff), then the alkalinity concentration will stay the same through time. If we divide this constant alkalinity concentration with the increasing runoff, we will get a decreasing normalized alkalinity concentration through time. But in reality, if we talk about the real weathering fluxes, it should be increasing, not decreasing because more material is released from the rock. Suppose the weathering cannot keep up with the runoff (i.e., the absolute amount of elements released from rocks is constant through time while runoff is increasing). In that case, we will obtain a decreasing alkalinity concentration and also a decreasing normalized alkalinity concentration. But in reality, the weathering flux itself should be constant, not decreasing. Hence, I am still not sure why the author picks this normalized value, as it simply could not represent real weathering behavior. I appreciate the author's response that "to study the change in alkalinity flux due to climate change, we considered runoff again". But my question here is that even if we can make a good prediction of normalized alkalinity concentration, it seems not reasonable to use this to represent weathering flux (or behavior). Thus, the feature importance in explaining this prediction might not be the same feature importance for explaining the weathering flux itself. I don't know why the author doesn't choose the weathering flux itself (which focuses on the real weathered amount and also considers runoff). Is it possible for the author to conduct a scenario test for using alkalinity flux (mol/km²/yr)?

2) Similar concerns also go to why the author uses catchment area as a predictor variable. I understand that: $Erosion = kAmSn$ (response to my original comment 13) but this area effect is more related to stream processes, not the erosion on land (see Line 181 in the text). With a higher slope, average erosion tends to be bigger. But with a larger area, I don't think the average erosion rate will necessarily increase. Since the author has all the data (i.e., erosion, A, and S), I suggest the author can plot Erosion vs. area and see whether erosion is in a positive correlation with A. Now back to using A as a predictor for normalized alkalinity flux. I see that adding A to the model increase the prediction power, but I still have a hard time understanding this from the mechanistic perspective (because we are dealing with normalized concentration, not the absolute amount of alkalinity).

3) Regarding original comments 8 and 12, and the new text in Line 149-171. I see two factors influencing the normalized alkalinity in the river. One is the soil pCO₂. The author states that with the increase in elevation, there will be a change in vegetation, climate, and soil properties (a lower CO₂ content). Hence, weathering flux will decrease due to low acid availability. The second is the river pCO₂ itself (river acidity). The author states that high elevation is linked to high erosion and high slope (which I agree with), which will lead to more turbulent flow in rivers, and thus a strong CO₂ degassing and carbonate precipitation, which will decrease the normalized alkalinity. However, as I see it, the second factor just leads to an underestimate of the alkalinity flux derived from weathering on land. Weathering on land is different from reaction in river. If we use the river alkalinity to measure the land weathering, then carbonate precipitation in the river

will decrease the “real” alkalinity derived from land. Therefore, even if we see a decrease in alkalinity with higher elevation, we might not be able to see the land weathering itself is decreasing with elevation.

4) Line 706-713: Where are the values given by the leave-one-out cross-validation for M1 to M5? A table like extended table 2 is needed here. In addition, I didn’t quite understand why the author says “Accordingly, M5 performed better than M4 at the outer parts of our temperature range.”? Sorry if I miss something. What are the outer parts of our temperature range? Without the leave-one-out table for the testing dataset, it is hard to understand what the author means.

5) Line 651-Line 658: I am not sure if I understand what the author tries to convey here. Is the author testing the influence of dam on normalized alkalinity signal? Or the influence of dam on ^{10}Be -derived erosion rates? “The variable proved to be significant but did not change the general trend in normalized alkalinity as a function of ^{10}Be -derived erosion rate.” Does this variable in this sentence mean “dam” itself or the area proportion of the watersheds influenced by the dam construction? If I understand correctly, dam will change the ^{10}Be -derived erosion rate and dam itself will also change the alkalinity flux. So if alkalinity is found to be strongly correlated with both dam and ^{10}Be -derived erosion rate, then it will be difficult for us to say it is the erosion (instead of dam) that is driving the change of alkalinity.

Other minor comments.

Regarding original comment 14, Yes. Soil regolith might be the better one (similar to Deng 2022 NC paper).

Regarding original comment 20, Yes. I would recommend the author use “)” instead of “[”. But I guess either is fine.

Line 29: is this increase ($0.5 \text{ t a}^{-1} \text{ km}^{-2}$) also for the mid-latitude regions? Or for the global land surface? I suggest the author makes this clearer.

Line 236: why minor influence?

Line 561-562: How does the author determine the area proportion in each watershed that is influenced by dam? Overlay the vector file from GOODD on the delineated watershed and calculate the proportion?

Authors' Response to Reviewer's Comments

#	Reviewer's Comment	Authors' Response	Position in Manuscript (Simple Mark-up)
1	In several responses, the author told the reviewer to see the provided R-script. But I am sorry that I didn't find any R-script in the uploaded files.	We have set up a Github link (https://github.com/nelelehmann/Alkalinity-responses-to-climate-warming-destabilise-the-Earth-s-thermostat/) where the R script can be accessed.	p. 34, l. 25 ff. and p. 36, l. 2 ff.
2	I am still not convinced about using the normalized alkalinity concentration to represent the weathering behavior. Here is one thought experiment. Imagine a single watershed with a time-series set of data. Assuming the runoff is changing from small to big over time. If the weathering can keep up with the runoff (for example, we see a linear relationship between the absolute amount of elements released from rocks and the runoff), then the alkalinity concentration will stay the same through time. If we divide this constant alkalinity concentration with the increasing runoff, we will get a decreasing normalized alkalinity concentration through time. But in reality, if we talk about the real weathering fluxes, it should be increasing, not decreasing because more material is released from the rock. Suppose the weathering cannot keep up with the runoff (i.e., the absolute amount of elements released from rocks is constant through time while runoff is increasing). In that case, we will obtain a decreasing alkalinity concentration and also a decreasing normalized alkalinity concentration. But in reality, the weathering flux itself should be constant, not decreasing. Hence, I am still not sure why the author picks this normalized value, as it simply could not represent real weathering behavior. I appreciate the author's response that "to study the change in alkalinity flux due to climate change, we considered runoff again". But my question here is that even if we can make a good prediction of normalized alkalinity concentration, it	Dear Reviewer, dear Editor, We think that we are in general agreement with the reviewer's points. We had chosen to use the normalized alkalinity in order to be able to perform a dilution correction, appreciating the effects of surface runoff (intense rain events, snow melt), which we consider as alkalinity free, as it is not impacted by weathering processes. If we had not done this, the alkalinity study would have been obscured by the influence of such surface runoff. Accordingly, we used the normalization as an analytical tool, the later quantitative integration does consider alkalinity concentration and actual runoff to establish alkalinity transports. If we take up the scenario of increasing runoff suggested by the reviewer, we see two possible cases: 1) Increased surface runoff leads to a dilution of the alkalinity concentration (pure rainwater or snow melt dilution effect) and accordingly also to a lowered normalized alkalinity. However, since we eventually multiply by runoff and by discharge (squaring the volume part), this dilution effect cancels out. The resulting alkalinity transport remains the same. 2) In this case, increased surface runoff causes a washout of groundwater containing alkalinity from reservoirs, so it does not cause a pure dilution of the alkalinity concentration, but may even cause a (temporary) increase in alkalinity concentration. Whether this will lead to a lower or higher normalized alkalinity,	R-script; p. 28, l. 22 ff.; p. 8, l. 9 ff.; p. 34, l. 17 ff.

	seems not reasonable to use this to represent weathering flux (or behavior). Thus, the feature importance in explaining this prediction might not be the same feature importance for explaining the weathering flux itself. I don't know why the author doesn't choose the weathering flux itself (which focuses on the real weathered amount and also considers runoff). Is it possible for the author to conduct a scenario test for using alkalinity flux (mol/km2/yr)?	depends on the ratio of the changes in alkalinity and runoff, respectively. Independently of that ratio, the later integration (multiplication with the square of the runoff as described in case 1) will yield enhanced alkalinity transports, since in contrast to case 1, here in case 2 extra alkalinity has been added to the runoff. We agree that the question whether alkalinity increases with runoff or not is one of the most crucial questions in Earth Sciences. However, we are not aware of any evidence in the literature indicating that an increase in runoff leads to a permanent increase in alkalinity. Rather, we think that after an alkalinity reservoir is washed out by a rain/snow melt event, the reservoir is depleted, resulting in a reduced concentration after the event. Thus, alkalinity generation is potentially not transport-limited, but the weathering itself, the chemical dissolution, is the limiting factor, i.e., kinetic control is exerted by chemical breakdown of the rock. This debate seems to be the “golden question” in the alkalinity issue. We appreciate the stimulating discussion and have therefore included a section “Some considerations on normalized alkalinity” on this topic in the Methods. Further, we see what the reviewer is trying to point out here (the feature importance for explaining normalized alkalinity may not be the same as for explaining alkalinity flux). We therefore developed a GAM for alkalinity flux (alkalinity concentration x runoff) to test whether all covariates included in our model M5 for normalized alkalinity are also significant predictors for alkalinity flux. Indeed, they are all significant predictors (included as smooth functions) at a significance level of 0.001. Here, we chose a GAM (with the covariates as smooth functions) instead of a GLM with the intention of using a fully data-driven approach to test whether these covariates are also significant when considered for alkalinity flux instead of normalized alkalinity. The
--	---	---

		shapes of the smooth functions of the individual covariates in the GAM (for alkalinity flux) are similar to the functional relationships of the individual covariates in the GLM (i.e., model M5 for normalized alkalinity), but not identical (see R script). We included a sub-clause in the main text noting that the five covariates included in model M5 for normalized alkalinity are also significant predictors in a GAM for alkalinity flux and referenced the Methods. We have added a corresponding paragraph in the methods. The permutation feature importance test revealed that the areal carbonate proportion is the most important covariate in the GAM for predicting alkalinity flux. This is consistent with the result of the permutation test for M5 predicting normalized alkalinity. The only difference in the results of the permutation test between the GLM (normalized alkalinity) and the GAM (alkalinity flux) is that erosion rate is ranked before MAT in the former, and after it in the latter (see R-script).	
3	Similar concerns also go to why the author uses catchment area as a predictor variable. I understand that: Erosion = $kAmSn$ (response to my original comment 13) but this area effect is more related to stream processes, not the erosion on land (see Line 181 in the text). With a higher slope, average erosion tends to be bigger. But with a larger area, I don't think the average erosion rate will necessarily increase. Since the author has all the data (i.e., erosion, A, and S), I suggest the author can plot Erosion vs. area and see whether erosion is in a positive correlation with A. Now back to using A as a predictor for normalized alkalinity flux. I see that adding A to the model increase the prediction power, but I still have a hard time understanding this from the mechanistic perspective (because we are dealing with normalized concentration, not the absolute amount of alkalinity).	 • Plot of $\log(\text{erosion rate})$ vs $\log(\text{catchment area})$: p-value = 0.08248; $R^2 = 0.013$; adjusted $R^2 = 0.0087 \rightarrow$ correlation is not significant at the 0.05 significance level • Plot of $\log(\text{erosion rate})$ vs. slope: p-value < 2.2×10^{-16}; $R^2 = 0.67$; adjusted $R^2 = 0.67$; positive correlation; correlation is significant at the 0.05 significance level • Plot of $\log(\text{erosion rate})$ vs. $(\log(\text{catchment area}) * \text{slope}) \rightarrow$ p-value value < 2.2×10^{-16}; $R^2 = 0.72$; adjusted $R^2 = 0.71$; positive correlation of erosion rate and the product of $\log(\text{catchment area})$ and slope \rightarrow Although $\log(\text{catchment area})$ by itself is not a significant predictor at the 0.05 significance level in the LM (linear model) of $\log(\text{erosion rate})$ and $(\log(\text{catchment area}) * \text{slope})$, the product of $\log(\text{catchment area})$ and slope is significant at the 0.05 significance level. Furthermore, the adjusted R^2 of this LM is greater than the one of the LM that only includes slope. 	R-Script

		Therefore, including catchment area in the product of $(\log(\text{catchment area}) \cdot \text{slope})$ improves the prediction power for erosion rate. Regarding the mechanistic perspective: We believe that a larger catchment area is associated with a higher normalized alkalinity by providing a greater opportunity for weathering and soil development (greater areas of finer-grained unconsolidated landforms which are ideal for weathering).	
4	Regarding original comments 8 and 12, and the new text in Line 149-171. I see two factors influencing the normalized alkalinity in the river. One is the soil pCO₂. The author states that with the increase in elevation, there will be a change in vegetation, climate, and soil properties (a lower CO₂ content). Hence, weathering flux will decrease due to low acid availability. The second is the river pCO₂ itself (river acidity). The author states that high elevation is linked to high erosion and high slope (which I agree with), which will lead to more turbulent flow in rivers, and thus a strong CO₂ degassing and carbonate precipitation, which will decrease the normalized alkalinity. However, as I see it, the second factor just leads to an underestimate of the alkalinity flux derived from weathering on land. Weathering on land is different from reaction in river. If we use the river alkalinity to measure the land weathering, then carbonate precipitation in the river will decrease the “real” alkalinity derived from land. Therefore, even if we see a decrease in alkalinity with higher elevation, we might not be able to see the land weathering itself is decreasing with elevation.	Very good point. We agree with the reviewer that a decrease in normalized alkalinity in the river due to strong CO₂ degassing and carbonate precipitation should not be equated with a decrease in weathering on land / in the soil. To clarify this distinction in the manuscript, we have added "riverine normalized" to the term "alkalinity concentration" so that it is clear that the degassing process in rapidly eroding catchments interacts only with alkalinity in rivers and not with the weathering (alkalinity generation) process itself.	p. 7, l. 23
5	Line 706-713: Where are the values given by the leave-one-out cross-validation for M1 to M5? A table like extended table 2 is needed here. In addition, I didn't quite understand why the author says "Accordingly, M5 performed better than M4 at the outer parts of our temperature range."? Sorry if I miss something. What are the outer parts of our temperature range? Without the leave-one-out table for the testing dataset, it is hard to understand what the author means.	We included the values given by the leave-one-out cross-validation (mean squared prediction errors) for models M1 to M5 in the Extended Data Table 1. We also added an Extended Data Table 3 (similar to Extended Data Table 2), which shows the mean squared prediction errors for the different temperature bands for models M3 to M5. At the outer parts of our temperature range, i.e., [-3,0) and [20,27) °C, M5 performed	p. 57 and p. 59

		better than M4. In the main text, we added references to the newly added column or table.	
6	Line 651-Line 658: I am not sure if I understand what the author tries to convey here. Is the author testing the influence of dam on normalized alkalinity signal? Or the influence of dam on 10Be-derived erosion rates? “The variable proved to be significant but did not change the general trend in normalized alkalinity as a function of 10Be-derived erosion rate.” Does this variable in this sentence mean “dam” itself or the area proportion of the watersheds influenced by the dam construction? If I understand correctly, dam will change the 10Be-derived erosion rate and dam itself will also change the alkalinity flux. So if alkalinity is found to be strongly correlated with both dam and 10Be-derived erosion rate, then it will be difficult for us to say it is the erosion (instead of dam) that is driving the change of alkalinity.	This variable in this sentence means the area proportion of the watersheds influenced by the dam construction. We edited the sentence to make this clearer. Moreover, we added a figure (Extended Data Fig. 15) to demonstrate that the addition of this variable (area proportion of the watersheds influenced by the dam construction) to model M5 does not change the general trend of normalized alkalinity as a function of the other covariates of model M5. Therefore, although we included this dam variable in the model M5, the ¹⁰ Be-derived erosion rate covariate is still a significant predictor for normalized alkalinity (with the same general trend as in M5 without the dam variable).	R-script; p. 31, l. 4 ff. and p. 55
7	Regarding original comment 14, Yes. Soil regolith might be the better one (similar to Deng 2022 NC paper).	Okay, we changed soil thickness to soil regolith thickness everywhere and edited Extended Data Fig. 2, 5, and 11 accordingly.	p. 40, p. 43, and p. 51
8	Regarding original comment 20, Yes. I would recommend the author use “)” instead of “[”. But I guess either is fine.	We changed [to).	p. 33, l. 8 ff. and p. 58
9	Line 29: is this increase (0.5 t a-1 km-2) also for the mid-latitude regions? Or for the global land surface? I suggest the author makes this clearer.	Done.	p. 2, l. 4
10	Line 236: why minor influence?	That was an error, we corrected the sentence accordingly.	p. 11, l. 4 ff.
11	Line 561-562: How does the author determine the area proportion in each watershed that is influenced by dam? Overlay the vector file from GOODD on the delineated watershed and calculate the proportion?	Yes. We only did this if the dam construction (information taken from the point vector file) was upstream from the sampling location.	

Reviewer #2 (Remarks to the Author):

I am happy to see the authors have addressed my concerns and made the text more polished. I really like all the discussion processes. I think the manuscript is ready to be accepted.

Authors' Response to Reviewer's Comments

#	Reviewer's Comment	Authors' Response	Position in Manuscript (Simple Mark-up)
1	I am happy to see the authors have addressed my concerns and made the text more polished. I really like all the discussion processes. I think the manuscript is ready to be accepted.	Great, we are happy to hear that. Thank you very much for the fruitful discussion.